# CLOUD: A Scalable and Physics-Informed Foundation Model for Crystal Representation Learning

Changwen Xu [1], Shang Zhu [1] & Venkatasubramanian Viswanathan [1,2] ✉

Predicting crystal properties is essential for understanding structure-property relationships and accelerating material discovery. However, conventional approaches like experimental measurements or density functional theory calculations are resource-intensive, limiting their scalability. While machine learning offers a promising alternative by learning complex structure-property relationships from data, existing models often rely on labeled data, adopt representations insufficiently capturing essential structural characteristics, and lack integration of physics, limiting their generalizability and interpretability. Here, we introduce CLOUD (Crystal Language mOdel for Unified and Differentiable materials modeling), a transformer-based framework trained on a Symmetry-Consistent Ordered Parameter Encoding (SCOPE) that encodes crystal symmetry, Wyckoff positions, and composition in a compact, coordinate-free string representation. Pre-trained on over six million crystals, CLOUD is fine-tuned on downstream tasks and achieves competitive performance across diverse material properties, demonstrating strong scaling with respect to data and model size. Furthermore, as a proof-of-concept of differentiable materials modeling, CLOUD is applied to predict the phonon-related properties by integrating with the Debye model. This approach enforces thermodynamic consistency and enables temperature-dependent property prediction without requiring additional data. These results demonstrate CLOUD's potential as a scalable and physics-informed foundation model for crystalline materials, unifying symmetry-consistent representations with physics-grounded learning for property prediction and materials discovery.

Predicting the properties of crystalline materials is of scientific and technological significance, as it enables insight into structure-property relationships and facilitates the discovery of functional materials[1-5]. Accurate knowledge of material properties underpins the design of functional materials across a broad range of applications, including electronics[6], energy storage[7,8], and catalysis[8,9]. One illustrative example is the heat capacity at constant volume ($C_v$), which serves as a representative example of a thermodynamic property that plays a key role in

vibrational analysis and phase stability[10-12]. As such, the ability to predict these properties from a material's structure is essential for accelerating the development of new materials and deepening our understanding of structure-property relationships.

Conventionally, crystal properties are evaluated through experimental characterization or theoretical simulations. In the lab, properties such as crystal structure, conductivity, and stability are measured using techniques such as X-ray diffraction, spectroscopy, and

[1]Department of Mechanical Engineering, University of Michigan, Ann Arbor, USA. [2]Department of Aerospace Engineering, University of Michigan, Ann Arbor, USA. ✉e-mail: venkvis@umich.edu

electrochemical analysis. On the theoretical side, Density Functional Theory (DFT) is widely used to compute formation energies, electronic structures, and elastic constants from first principles. However, these approaches are often time-consuming and resource-intensive, limiting their scalability for high-throughput screening and materials discovery[13,14]. Consequently, materials development has been constrained to a limited portion of the vast chemical space of synthesizable crystalline materials due to the high costs associated with evaluating material properties within specific systems[15,16]. To overcome these limitations, the materials science community has increasingly embraced data-driven methods for predicting material properties with high throughput, reducing reliance on costly simulations and resource-intensive experiments. Among these, machine learning surrogates have proven especially effective, as they can learn complex structure-property relationships from data and enable rapid, scalable property predictions. Leveraging these strengths, ML models have the potential to substantially broaden the scope of material performance evaluation in the vast chemical space of crystals[17]. Pioneering work has significantly improved the prediction accuracy for crystals[18–23]. Rational design of representations that map crystals to a continuous vector space has enabled the development of more accurate and generalizable machine learning models for predicting crystal properties. Early models focused on composition-based representations[20,24,25], which are simple and broadly accessible but suffer from limited expressiveness because they cannot distinguish between structural polymorphs. Structure-based models overcome this limitation by leveraging atomic coordinates, often through graph neural networks, to capture geometric and spatial information[18,19,21,26,27]. However, these models struggle with long-range interactions[28] and require costly structural data. To bridge the gap, coordinate-free models have emerged, offering a balance between structural awareness and computational efficiency by encoding symmetry and connectivity without explicit coordinates[29–31]. Nevertheless, current coordinate-free representations still face challenges in efficiently encoding critical structural information for accurate property predictions, motivating the need for improved representations that balance structural richness and computational simplicity.

Despite the advances in material property prediction with machine learning, state-of-the-art (SOTA) models typically rely on labeled training data obtained through costly wet-lab experiments or DFT calculations. This reliance limits their applicability and scalability due to the scarcity, high cost, and variability of labeled materials datasets[32]. Moreover, these models often exhibit poor out-of-distribution (OOD) generalization, which is critical for real-world applications where crystals with novel compositions or structures are common[17]. Recently, foundation models (FMs)[33], which are trained on large-scale, diverse data and can be adapted to a wide range of downstream tasks, have emerged as a promising solution. Most FMs adopt a transformer-based architecture[34], enabling them to be pretrained at scale using vast amounts of unlabeled data[35–37], which are orders of magnitude more abundant than labeled data. The key motivation for pre-training on large-scale unlabeled datasets is to learn universal representations of crystals through self-supervised learning. Aside from language-model-style pre-training, alternative self-supervised strategies have been explored in the crystal domain, such as Crystal Denoising Self-Supervised Learning (CDSSL)[38], which pre-trains graph neural networks by perturbing atomic positions and learning to recover the original interatomic distances. These representations, whether learned from sequence-based or geometric-denoising objectives, can be effectively transferred to a wide range of downstream tasks, improving model performance even in low-data regimes. Furthermore, the implicit parallelism of attention allows for massively scalable transformer architectures, whose performance can improve predictably with the size of the model and training data, following neural scaling laws[39–41]. If such a scaling behavior holds, FMs are expected to achieve SOTA performance simply by increasing the data and model capacity. FMs leverage self-supervised pretraining to capture high-level structural and chemical patterns from unlabeled data[33,42], which has shown promise in predicting crystal properties[43–45]. In the molecular domain, models like ChemBERTa[46] and MoLFormer[47] have demonstrated that performance improves with larger pretraining datasets across both regression and classification tasks. Systematic studies on neural scaling laws have been conducted for molecular property prediction[42,48], yet such analyses remain largely unexplored for materials-focused foundation models.

Another key question of developing foundation models for crystalline materials lies in the incorporation of physics. Even though high predictive accuracy can be achieved with machine-learning models, the consistency of predictions in physics is not necessarily guaranteed. For example, previous work[28,49] that learns the mapping from crystal structures to heat capacity ($C_v$) in a pure data-driven manner fails to learn the temperature dependency of $C_v$, hence these models are only capable of predicting $C_v$ values under exactly the same temperature as the training data, which limits their applications in real-world material science tasks. Gurunathan et al.[50] predicted the phonon density of states (pDOS) from which $C_v$ could be calculated with the standard Bose-Einstein distribution formalism, accomplishing enhanced predictive accuracy for temperature-dependent phonon-based properties. However, the model itself is still predicting the target property in one shot, given the input structure without injecting physics prior. The machine learning model can be further integrated with physical laws by predicting the variables within these laws, which, in turn, determine the target property. This framework can be trained end-to-end, provided that the physical law is differentiable. Recently, DiffMix[51] has been developed as a differentiable geometric deep learning framework for mixtures that learns physical coefficients from predefined mixture physics laws, demonstrating promising performance compared to purely data-driven methods, especially on small datasets. DiffMix exhibits strong extrapolation to higher temperatures, enhancing predictive reliability beyond training conditions. By integrating physics-based modeling with geometric deep learning, it enables more accurate and generalizable predictions, supporting efficient material design and optimization. Recent advances in differentiable materials modeling[52–54] have unified physical simulation and learning by enabling gradient-based optimization through physical laws, such as thermodynamic potentials, finite element solvers, or lattice mechanics. These frameworks enable models to predict intermediate physical quantities and to propagate gradients through differentiable physical pipelines, supporting data-efficient, generalizable material design.

Herein, we design Symmetry-Consistent Ordered Parameter Encoding (SCOPE) to represent crystal structures as sequences and consequently propose Crystal Language mOdel for Unified and Differentiable materials modeling (CLOUD), a transformer-based foundation model for accurate and generalizable crystal property predictions. The model consists of a BERT[35] encoder and a multi-layer perceptron (MLP) prediction head. SCOPE is a symmetry-consistent string representation for crystals that integrates symmetry, equivalent sites, and compositions in a coordinate-free encoding of the crystal structures, as shown in Fig. 1b. Using this representation, CLOUD is pre-trained via masked language modeling (MLM) on ~6.3M unique crystals collected from the OPTIMADE[55], representing one of the largest high-quality datasets of DFT-relaxed crystals to date, then fine-tuned on downstream datasets to learn task-specific structure-property relationships (Fig. 1c). CLOUD exhibits competitive model performance when evaluated for predicting various DFT-calculated material properties, indicating its potential as the surrogate model in the loop of material discovery. Additionally, we investigate the scaling performance of CLOUD, suggesting potential performance gains with further scaling of data and model size. We further integrate CLOUD in a differentiable physics framework for physics-consistent property predictions. To showcase this capability, we fuse the

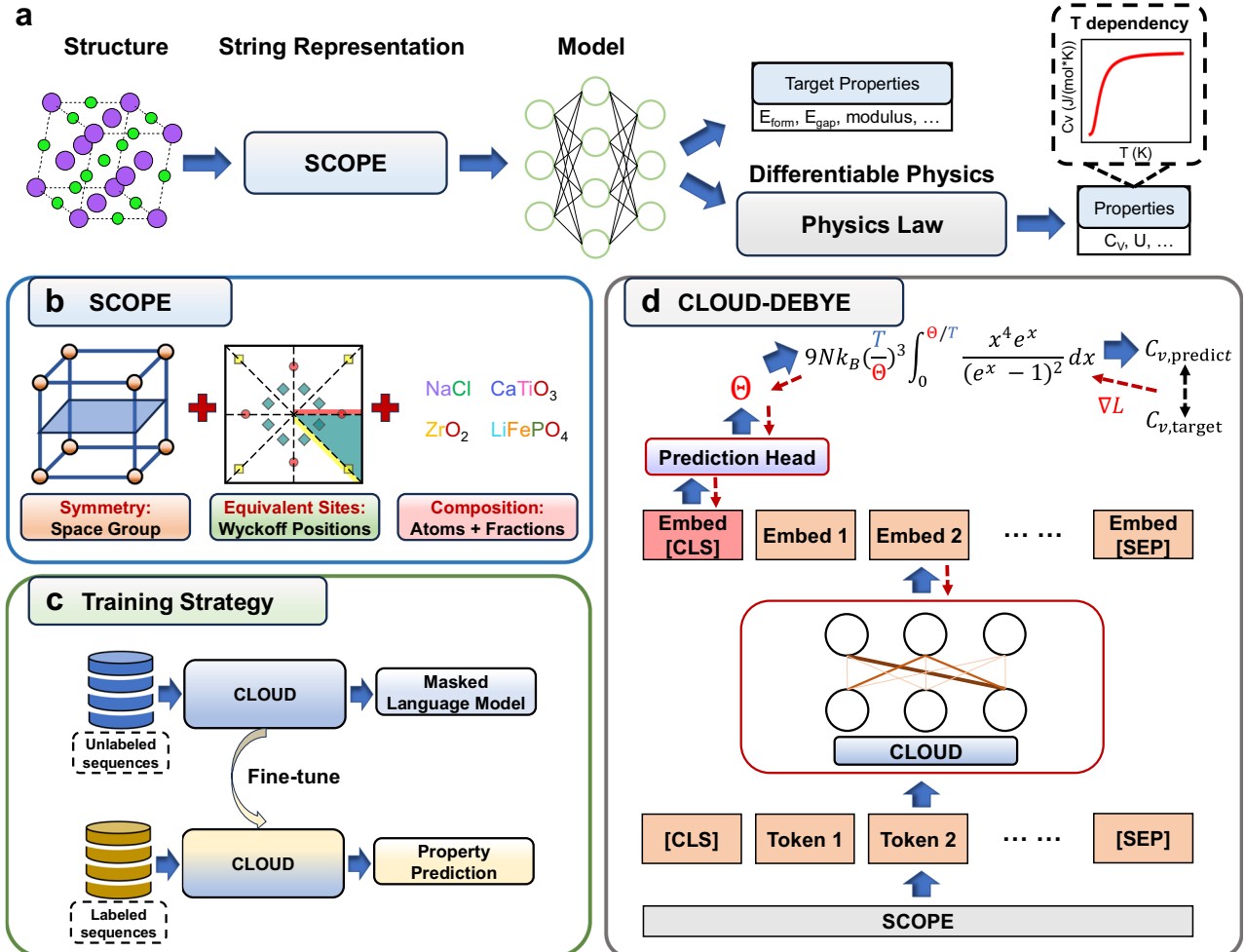

**Fig. 1 | Overview of CLOUD. a** For a given crystal structure (the example structure is plotted with ASE[106]), we build the string representation SCOPE for it, which will serve as the input of the deep learning model CLOUD which predicts the material properties. Aside from directly outputting the properties of interest, the model can also predict physical variables that satisfy known physical laws, enabling physics-consistent property predictions. The three blocks below illustrate the key components of the CLOUD framework. **b** SCOPE representation, a symmetry-consistent string representation for crystal structures, consists of space group generator strings, Wyckoff position symbols, and material composition. **c** Our model CLOUD is pre-trained on a million-scale unlabeled data with MLM to recover the original tokens, while the feature vector corresponding to token `[CLS]' from the last hidden layer is used for prediction when fine-tuning on diverse downstream tasks (labeled material property data). **d** CLOUD-DEBYE is an example of integrating the foundation model CLOUD and physics laws in a differentiable physics framework. The example exhibits the prediction for constant-volume heat capacity ($C_v$), where CLOUD outputs the Debye temperature ($\Theta$) and the Debye model predicts $C_v$ using $\Theta$ as input under different temperatures. The differentiable implementation of the Debye model enables training the model end-to-end with $C_v$ targets.

model with Debye model[56] (CLOUD-DEBYE, Fig. 1d) and assess its ability to predict phonon internal energy and heat capacity, two properties that rely on long-range interactions of crystal structures. CLOUD-DEBYE exhibits better predictive performance than CLOUD itself and other GNN or GNN-hybrid models; at the same time, its predictions for the heat capacity across different temperatures match the experimental data and ensure thermodynamic consistency. Such results suggest the potential of CLOUD to learn long-range interactions in crystal structures and to verify physical consistency by integrating with physics-based models in materials applications. Overall, our model extends machine learning for crystal representation learning and property prediction by developing a string representation for effective encoding, building an accurate, scalable foundation model, and unifying physics models and data-driven techniques, thereby enabling the potential to boost real-world material design and discovery.

## Results

### SCOPE crystal representation

Crystal representations in machine learning often fall into two broad categories: composition-based and structure-based. Composition-only representations, while easy to obtain, lack the specificity required to uniquely determine a structure due to the non-uniqueness of composition-structure relationships. On the other hand, structure-based graph representations typically rely on 3D atomic coordinates, which are computationally intensive and often unavailable. Moreover, both categories often overlook a fundamental feature of crystals: symmetry, which governs many physical properties such as electronic behavior, optical response, and mechanical stability[57]. To capture symmetry explicitly, many recent models incorporate equivariance constraints[58–60], although these introduce complexity into the architecture. Huang et al.[31] took a step forward by incorporating space group numbers and symbols into a sequence-based representation suitable for language models. Meanwhile, recent work has demonstrated the importance of explicitly incorporating crystallographic symmetry and Wyckoff information into generative and predictive models. For example, WyCryst[61] introduces a Wyckoff-based representation that preserves symmetry during generation and relaxation; WyFormer[62] encodes crystals as fused element-Wyckoff tokens to enable symmetry-conditioned autoregressive generation; SymmCD[63] decomposes crystals into asymmetric units and symmetry operations, ensuring symmetry-preserving diffusion even for

rare space groups. These works underscore that symmetry-aware, Wyckoff-based encodings serve as effective and expressive representations for crystal structures, substantially improving the physical validity and diversity of generated crystals. However, limitations exist in existing symmetry-based representations for crystals. For example, the symbolic identifiers used in Huang et al.[31]—such as $P2_1/c$ or #14 − are compact summaries rather than full encodings of symmetry operations. They do not convey the complete set of transformations that define the space group, such as mirror planes, rotation axes, or screw axes, nor do they capture the relationships between different space groups.

To address these limitations, we propose SCOPE, a coordinate-free string representation that integrates three essential components of a crystal structure: the space group, the Wyckoff positions, and the composition. By omitting explicit atomic coordinates and instead encoding symmetry constraints and atomic assignments, SCOPE captures the essential building blocks of a crystal structure in a compact and expressive form.

A crystal structure can be described mathematically by its lattice and basis. Following the notations in Jiao et al.[64], the lattice is a periodic grid in three-dimensional space defined by three linearly independent lattice vectors forming a matrix $L = [l_1, l_2, l_3] \in \mathbb{R}^{3 \times 3}$. Atomic positions within a unit cell are specified by fractional coordinates $F = [f_1, f_2, \ldots, f_N] \in [0, 1)^{3 \times N}$, which relate to Cartesian coordinates by $X = LF$. A crystal is fully specified by its lattice $L$, the set of atomic positions $X$, and the chemical species occupying those positions. However, not all configurations are physically meaningful−most crystals exhibit symmetry, which constrains how atoms can be arranged.

The full symmetry of a crystal is captured by its space group, a finite set of symmetry operations $g \in G$ that map the crystal onto itself. Each operation acts on the atomic coordinates $X \in \mathbb{R}^{3 \times N}$ as:

$$g \cdot X := OX + t1^\top \qquad (1)$$

where $O \in O(3)$ is an orthogonal matrix representing rotations, reflections, or rotoinversions, and $t \in \mathbb{R}^3$ is a translation vector. The crystal structure $\mathcal{M}$ is said to be invariant under $g$ if $g \cdot \mathcal{M} = \mathcal{M}$ (here, "=" denotes equivariance under the group action).

Traditionally, space groups are denoted by Hermann-Mauguin symbols or numbers (1-230), which provide compact, human-readable summaries of the symmetry elements. However, these representations are not sufficiently expressive for machine learning−they do not enumerate the full set of symmetry operations or distinguish subtle relationships between different groups.

To overcome this, SCOPE uses generator strings[65], which provide a symbolic representation of the symmetry generators for each space group. These generators are a minimal set of operations from which the entire space group can be constructed, allowing for a more granular and interpretable encoding of symmetry.

Among all the possible $g \in G$ in a space group, a generator is a fundamental symmetry operator from which all other operations of the space group can be derived through the combination of those basis operators. By selecting a minimal set of generators, we can fully describe the symmetry content of a space group. Therefore, each generator is represented with a symbolic notation to form the generator string for space groups. There are 14 generator matrices for all the symmetry operations besides translational symmetry (namely, $O$ in $g$), and these matrices are represented by letters ranging from $a$ to $n$. Combining the generator matrices with the translation components ($t$ in $g$), which are represented with 10 upper-case letters, we can obtain all the symmetry operators. Using the generators, it is possible to compile all 230 space groups into a short ASCII file of only 4104 bytes. The generator strings form a compact representation of all the allowable basic symmetries of the crystal structure, hence making the generator strings fingerprints of symmetries. The generator strings are particularly well-suited for language modeling, as they preserve the

symmetry operations and provide a richer input format than traditional number-based space group identifiers. More information on the generator strings can be found in the Methods section and Section 10.3.6 in De Graef et al.[65].

Following the symmetry part, SCOPE encodes the atom positions under the constraints of symmetry. Atom positions are usually represented by atomic coordinates in the form of $(x, y, z)$ in 3D space. However, using coordinate information is computationally intensive in terms of both data acquisition and model training. The acquisition of coordinate information necessitates DFT calculations for structure relaxation, while the model will grow extremely large as the crystal system grows larger. Alternatively, Wyckoff positions[29,66] describe sets of symmetry-equivalent points in the unit cell given space group could be used for atomic position representation. Each Wyckoff position corresponds to a class of points whose site-symmetry group is conjugate under $G$. The site-symmetry group of a point $f_i \in [0, 1)^3$ is defined as the stabilizer subgroup:

$$G_i = \{g \in G | g \cdot f_i \sim f_i\}, \qquad (2)$$

where $g \cdot f_i = Rf_i + t$, and ~ denotes equivalence under lattice translations. This group contains all symmetry operations that leave the point invariant up to translation.

Applying the full space group $G$ to a representative point $f_i$ generates a set of symmetry-equivalent positions:

$$\mathcal{W}_i = \{g \cdot f_i | g \in G\}, \qquad (3)$$

whose cardinality is called the *multiplicity* of the Wyckoff position. Each Wyckoff position is denoted by a label such as $4d$, where the number indicates the multiplicity $|\mathcal{W}_i|$, and the letter encodes the relative site-symmetry: labels earlier in the alphabet correspond to higher-symmetry positions.

The set of available Wyckoff positions is determined by the space group, and each atom in a crystal can be assigned to a Wyckoff position according to its symmetry environment. In our SCOPE representation, we encode the occupied Wyckoff positions and their corresponding chemical elements. This yields a compact and symmetry-consistent description of the crystal structure that is invariant under symmetry-preserving transformations and independent of explicit atomic coordinates.

We complete the representation by including the elements that make up the material, along with their fractions in the material. As a result, the representation is informed of the material's composition, ensuring that the model captures both the identity and proportion of elements.

Combining the three parts above, we obtain the symmetry-consistent string representation for crystal structures. Examples of representations can be found in Fig. 2a. The representation forms a top-down description of the crystal structure – starting from symmetry, which is global information of the structure, followed by the equivalent sites under the symmetry and the stoichiometry of the crystal – without explicitly encoding the structure itself, accomplishing efficient encoding of crystals. This provides a more granular and physically complete encoding than categorical site-symmetry tokens[62] or binary symmetry matrices[63]. Unlike WyCryst's one-hot representation with degrees of freedom[61], SCOPE captures the complete set of symmetry generators in a compact symbolic form, enabling fine-grained distinction between closely related space groups and naturally aligning with MLM for large-scale transformer pretraining. With the SCOPE representation for crystal structures as the input, we tokenize the sequence so that each chemically meaningful unit is treated as a single token. Specifically, each symbol in the generator string is tokenized individually, each Wyckoff position symbol is treated as a single token, and both element symbols and fractional values are

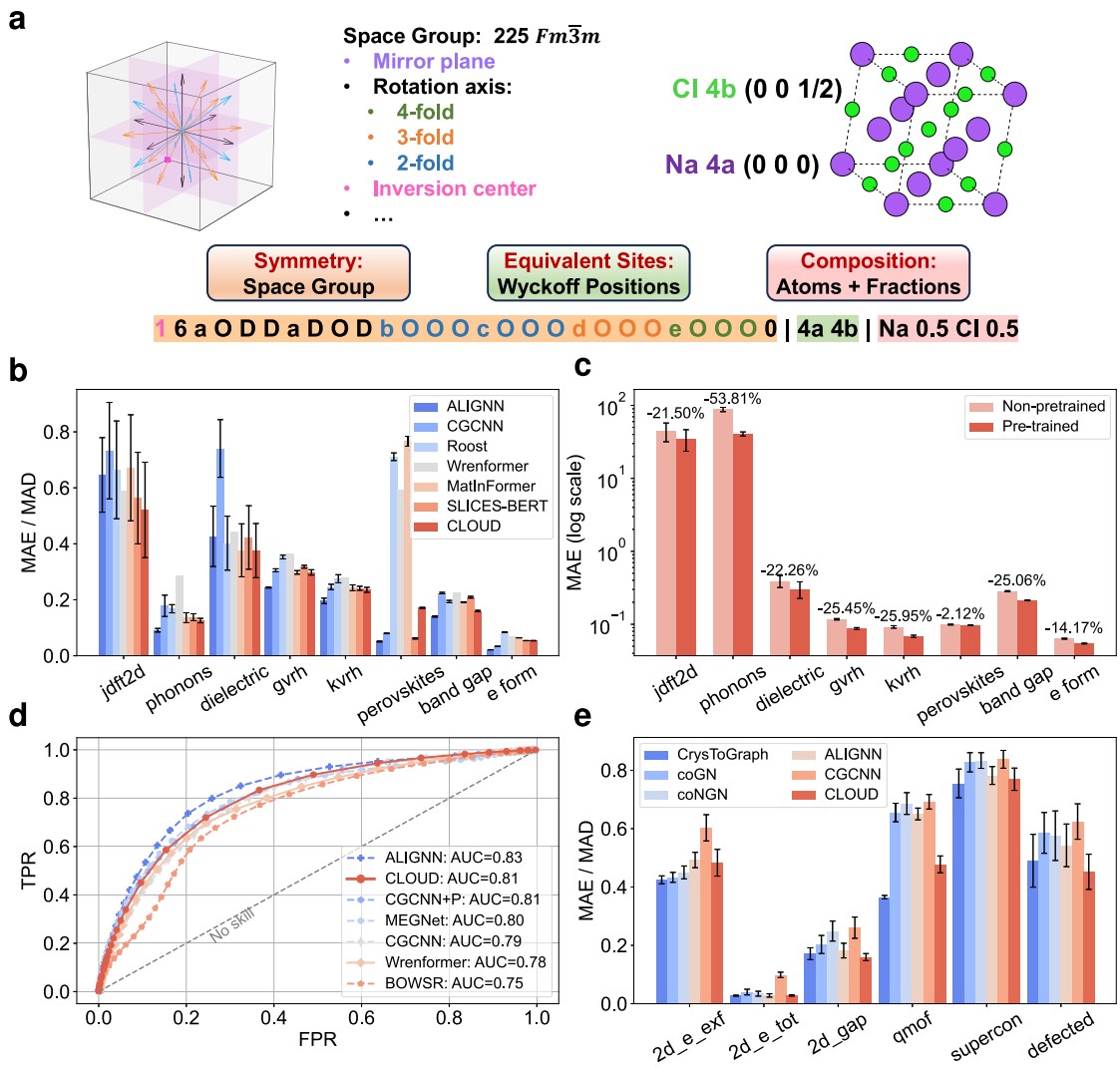

**Fig. 2 | Illustration of SCOPE representation and fine-tuning results of CLOUD. a** SCOPE consists of space group generator strings, Wyckoff position symbols, and material compositions. The example of NaCl (purple spheres represent Na atoms and green spheres represent Cl atoms) includes the illustration for space group symmetries and equivalent sites in the unit cell. **b** Fine-tuning results on MatBench regression tasks. All results are shown in the mean of test MAE/MAD in five-fold cross-validation, along with the standard deviation plotted as the error bars. **c** Comparison of five-fold average MAE results on MatBench datasets between CLOUD trained from scratch (non-pretrained) and pre-trained with MLM. The MAE results are shown in base-10 logarithm. The percentage reduction in MAE is annotated above each dataset, and the error bars are plotted. **d** Receiver operating characteristic (ROC) curves for models evaluated on MatBench Discovery for structure stability classification. False positive rate (FPR) and true positive rate (TPR) are the fraction of nonstable/stable structures classified as stable. Area under curve (AUC) scores are calculated and presented in descending order. **e** Fine-tuning results on UnconvBench general predictive tasks. All results are shown in the mean of test MAE/MAD in five-fold cross-validation, along with the standard deviation plotted as the error bars.

represented as distinct tokens. All the float numbers are rounded to 2 decimal places.

## Transferable and scalable crystal representation learning with CLOUD

CLOUD adopts the BERT architecture[35] and follows the pretrain-finetune paradigm for training. The transformer model is based on the attention mechanism[34], enabling it to effectively model long-range interactions and contextual relationships within sequences. Specifically, it employs scaled dot-product attention, which operates through query, key, and value matrices. Further details on attention are provided in the Methods section. In our implementation, the transformer encoder consists of 12 hidden layers, each comprising 12 attention heads. The hyperparameters of CLOUD are initially set based on standard BERT configurations and subsequently tuned to optimize model performance. The hyperparameters that are experimented with are summarized in Table S1. The transformer encoder is pretrained via

MLM[67–69] for effective representation learning from the unlabeled sequences, where 15% of the tokens in a sequence are randomly selected for potential replacement, and the pretraining objective is to reconstruct the original tokens by leveraging contextual information. The pretrained model is subsequently fine-tuned in a supervised manner for crystal property prediction. Specifically, the final hidden states (feature vector) of the special token '[CLS]', positioned at the beginning of the sequence, is passed through the prediction head to generate the prediction output.

We use the Open Databases Integration for Materials Design (OPTIMADE)[55] dataset for CLOUD pre-training. OPTIMADE comprehensively collects crystal data from various databases and provides an API for users with easy access to download the data. We download and de-duplicate the initial ~ 13M CIF files from OPTIMADE, resulting in ~ 6.3M data for pre-training. With the pre-trained model, we evaluate the predictive performance of CLOUD on MatBench datasets[70] which comprise diverse materials properties computed via DFT. We compare

CLOUD to diverse baseline models in terms of mean absolute error (MAE) normalized by mean absolute deviation (MAD), which is given by $\mathrm{MAE/MAD} = \frac{\sum |y_i - y_{i,\text{true}}|}{\sum |y_{i,\text{true}} - \bar{y}|}$ so that the metric is invariant to scaling, enabling comparison across datasets of different units[21]. The baseline models are primarily divided into three groups: structure-based models, which utilize coordinates of atoms as input: CGCNN[19] and ALIGNN[21], structure-agnostic models which do not require structural information: Roost[20], and coordinate-free models which rely on the structures to build the input while do not require the inclusion of atomic coordinates: Wrenformer[71], MatInFormer[31], and SLICES-BERT[30]. CLOUD achieves competitive results across nearly all eight regression tasks compared to structure-agnostic and coordinate-free models. As shown in Fig. 2b, CLOUD outperforms previous coordinate-free models in 7 out of 8 tasks and overall performs the best in jdft2d and dielectric datasets. Notably, the model attains strong performance even on data-limited tasks, suggesting that pre-training on large unlabeled datasets enhances generalization. Additionally, CLOUD surpasses MatInFormer[31], indicating that explicit symmetry encoding via generator strings contributes to improved predictive accuracy. Compared to SLICES[30], which encodes bond connectivity while neglecting overall symmetry, CLOUD reaches lower prediction error with a much more compact representation (the mean sequence lengths for CLOUD and SLICES are 31 and 513, respectively, for the same pre-training dataset), and consequently less computing budget. SLICES-BERT only surpasses CLOUD on perovskites on which none of the structure-agnostic or coordinate-free models perform well while SLICES benefits from connectivity information which conveys more subtle differences between perovskite structures. We provide the MAE results for each model on the eight regression tasks in Table S3. Fig. 2c presents the comparison between the results of CLOUD with/without pre-training. Model performance on downstream tasks is significantly better when pre-trained than when trained from scratch.

Aside from predicting the material properties from the relaxed crystal structures, we are also interested in assessing CLOUD's capability for structure stability prediction without knowing the relaxed crystal structures from DFT calculations. Therefore, we fine-tune the pre-trained model on MatBench Discovery[71] which is designed to evaluate machine learning models for materials discovery, particularly for predicting the thermodynamic stability of materials. The dataset aims to reflect practical challenges in the discovery process by requiring predictions based on unrelaxed crystal structures, which avoid reliance on expensive DFT calculations. We fine-tune CLOUD on Materials Project[72] formation energy training data and test on WBM dataset[73] which contains ~ 257K OOD crystal structures generated by systematically substituting elements in pre-existing structures from Materials Project. Model performance is evaluated in terms of the ROC-AUC score, which is calculated by varying the stability threshold, recording the false positive rate (FPR, the fraction of unstable structures being misclassified as stable) and true positive rate (TPR, the fraction of stable structures that are correctly identified), plotting the ROC (Receiver operating characteristic) curve, and calculating the area under curve (AUC) score. In contrast to directly predicting the formation energy for relaxed structures where CLOUD gives a larger prediction error than many structure-based GNNs, CLOUD manifests comparable classification performance despite not using atomic coordinates, given that the model achieves the AUC score of 0.81 which is close to that of ALIGNN[21] and higher than that of CGCNN[19] and MEGNet[74], as shown in Fig. 2d. These results highlight CLOUD's potential for accelerating materials discovery by effectively screening unrelaxed structures and directly predicting the stability of the relaxed ones, thereby minimizing reliance on costly DFT calculations. To further showcase the capability of CLOUD in material discovery, we conduct a prospective discovery-style evaluation. We use CLOUD's predicted formation energies to derive $E_{\text{hull}}$, rank candidates by predicted stability, and quantify discovery efficiency using Precision@k, Enrichment Factor, and Cost-to-$M$-stable metrics. These metrics emulate a realistic screening workflow under finite resource constraints. As shown in Figure S2a–c, CLOUD substantially improves efficiency by reducing the number of DFT calculations required compared to random screening. In a case study on the Al-Fe system, CLOUD successfully identifies $Al_2Fe$ as stable, despite its absence from the Materials Project convex hull, in agreement with results from the WBM dataset (Fig. S2d). These results highlight the strong potential of CLOUD to accelerate the discovery of novel stable materials.

Noticeably, the crystal structures from MatBench or Matbench Discovery datasets are 'regular' crystals that predominantly feature well-ordered structures and lack the structural diversity, dimensional variation, or disorder commonly seen in more complex or 'unconventional' materials, which are, in fact, 'conventional' in real-world conditions under finite temperatures. Therefore, we fine-tune the model on crystal structures from UnconvBench[75], which includes crystals with defects, large unit cells, or low-dimensional structures. Different from the cases for MatBench in which the sophisticated structure-based GNN models like coGN[22], coNGN[22], and ALIGNN[21] show superior accuracy, CLOUD demonstrates comparable performance to the SOTA model CrysToGraph[75] and significantly outperforms the other structure-based models (Fig. 2e). Such evidence reveals the promising potential of CLOUD to be applied for material discovery. In addition, the best model for each UnconvBench task is either CrysToGraph or CLOUD, both of which take into account global information about crystal structures. Distinct from CrysToGraph which learns the long-range interactions with a graph-wise transformer, CLOUD directly encodes symmetry information via generator strings and Wyckoff positions, providing an alternative for encoding global information for crystal structures. We provide the MAE results for each model on the six predictive tasks in Table S6.

To better understand how the SCOPE representation improves the modeling of crystal structures, we examine the attention scores between the [CLS] token and the other tokens. Attention scores indicate the correlation between tokens, thereby reflecting the model's interpretation of the encoded features. We focus on the attention scores between [CLS] and the other remaining tokens in the sequence because the [CLS] embedding is fed to the prediction head for property prediction, and close attention scores likely imply a significant contribution of the corresponding tokens to the material property prediction.

To illustrate, we compute attention scores across hidden layers and attention heads for crystal structures in the *gvrh* test set from MatBench. The attention scores between [CLS] and other tokens are aggregated across attention heads, from which we identify the top-$k$ important tokens with the highest attention scores for each data point. We prioritize analyzing the attention scores from the first hidden layer as it is closest to the input and consequently learns more semantic information from the sequences[76]. To quantify the relationship between [CLS] and space group tokens, we introduce two probabilities: $p_1$, the probability that at least one space group token appears in the top-$k$ list for a given sequence, and $p_2$, the proportion of space group tokens among the top-$k$ tokens. We vary $k$ and find that the probability of retrieving at least one space group token among the top-$k$ tokens ($p_1$) increases from 0.93 at $k = 1$ to 0.99 at $k = 3$, while the average proportion of space group tokens among the top-$k$ tokens ($p_2$) remains consistently high, ranging from 0.91 to 0.93. This confirms that space group tokens receive significant "attention" (contributions to the machine-learned feature), considering their average portion in a sequence in this dataset is ~ 0.64. To assess the statistical significance of this prioritization, we perform a one-sample t-test comparing the observed values of $p_1$ and $p_2$ to a baseline of 0.5, which serves as the null hypothesis representing random or uniform attention. The test rejects the null hypothesis ($p < 0.001$), indicating that $p_1$ and $p_2$ are

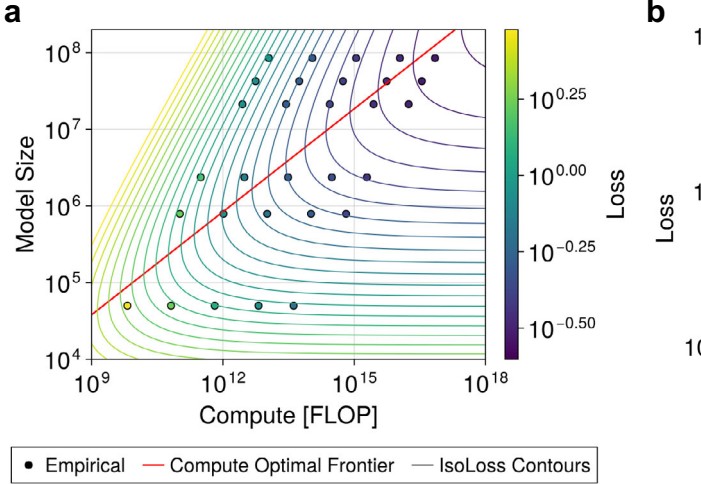

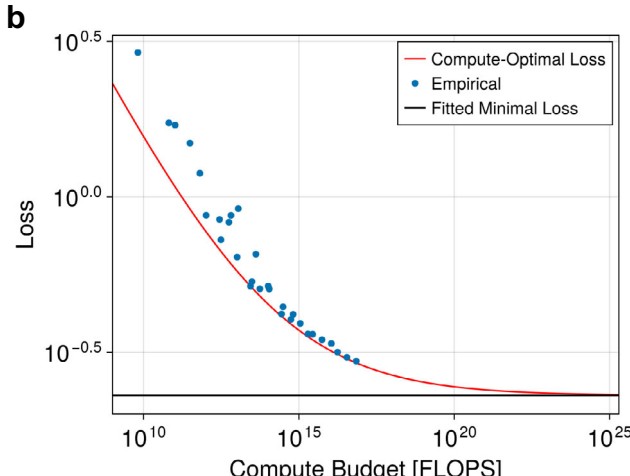

**Fig. 3 | Training compute optimal models for pre-training CLOUD.**
**a** Visualization of the scaling law fitted to empirical pre-training loss data. The black dots represent empirical observations, the red line indicates the compute-optimal frontier, and the contour lines represent iso-loss curves derived from the fitted

scaling function. **b** Pre-training loss as a function of total compute budget (FLOPs). The blue dots correspond to empirical results, while the red curve shows the compute-optimal loss predicted by the scaling law.

significantly greater than the expected 0.5. The 95% confidence intervals for $p_1$ ([0.9236, 0.9444] for $k = 1$, [0.9820, 0.9916] for $k = 2$, [0.9886, 0.9960] for $k = 3$) and $p_2$ ([0.9236, 0.9444], [0.9165, 0.9329], [0.9002, 0.9156]) consistently exceed 0.64, demonstrating that space group tokens are systematically prioritized. These results validate that the model effectively learns symmetry-consistent representations rather than distributing attention based solely on token frequency.

In addition to the evaluation of model prediction accuracy, it is crucial to understand how performance improves with increased model and data–an essential consideration for foundation models as they are often deployed at large scales to achieve generalization and transferability across tasks. Therefore, we further examine how the model performance scales with the number of pre-training data and model parameters. The model performance (cross-entropy loss for the pre-training task) empirically scales with the number of tokens $D$ and non-embedding model parameters $N$ (excluding the model parameters for vectorizing tokens) in a power law relationship, which is expressed by Hoffmann's scaling law[40]:

$$L(N, D) = \frac{A}{N^\alpha} + \frac{B}{D^\beta} + E \qquad (4)$$

We vary the number of pre-training data and non-embedding parameters in the encoder of the model to fit the scaling law accordingly. We solve the optimization problem with the L-BFGS algorithm[77] to obtain the following result:

$$A = 90.7464, B = 21.9854, E = 0.2296, \alpha = 0.4339, \beta = 0.3518$$

With the fitted scaling law, we estimate the optimal model size for a given compute budget. Following the derivation in literature[40], the optimal model size and data size can be written as:

$$N_{opt}(C) = G\left(\frac{C}{6}\right)^a, \quad D_{opt}(C) = G^{-1}\left(\frac{C}{6}\right)^b \qquad (5)$$

where

$$G = \left(\frac{\alpha A}{\beta B}\right)^{\frac{1}{\alpha+\beta}}, \quad a = \frac{\beta}{\alpha+\beta}, \quad b = \frac{\alpha}{\alpha+\beta} \qquad (6)$$

Given the $\alpha$ and $\beta$ we fitted, we can readily derive that $a = 0.45$ and $b = 0.55$ for CLOUD. Interestingly, the $a$ and $b$ from the empirical scaling law that we fit for CLOUD are close to the values in Hoffmann's scaling law, where $a = 0.46$ and $b = 0.54$ (derived from parametric modeling of the loss)[40]. Given $a \approx b$, the number of parameters and number of tokens should be scaled evenly, unlike the case for Kaplan's scaling law, which infers that the model size should scale faster than the number of tokens ($a = 0.73$, $b = 0.27$)[39].

The fitted scaling law is also visualized in Fig. 3. The fitted entropy term $E$, which reflects the natural entropy of text data, is equal to 0.2296 for CLOUD, significantly lower than the $E = 1.69$ reported in ref. 40, suggesting that the underlying data distribution for CLOUD is more structured or predictable, allowing for a smaller irreducible loss floor. The result aligns with our expectation, as the string representation that we design follows certain rules distilled from chemical knowledge.

Despite the strong scaling performance, the visualization in Fig. 3b, however, also reveals the limitation of scaling. The entropy term $E$, which reflects the natural entropy of the text data, the benefits of scaling diminish because the loss asymptotically approaches $E$ as the compute budget increases, highlighting that beyond a certain compute threshold, additional resources yield diminishing returns. This suggests that while scaling remains beneficial, achieving further gains may require innovations in architecture, optimization, or data efficiency rather than simply increasing model size and the number of training tokens. Specifically, our symmetry-consistent string representation primarily focuses on efficient encoding, though it does not ensure completeness, as it does not encode free variables in Wyckoff positions or lattice parameters for unit cells. Relevant discussions are also provided in section S1 in Supplementary Information. Therefore, future work that provides richer information while maintaining efficiency is required to design foundation models for crystal structures.

### CLOUD-DEBYE: Integration of CLOUD in a differentiable and physics-consistent materials thermodynamics framework

To advance beyond accurate but purely data-driven predictions, we propose integrating CLOUD with physical models to form a differentiable, physics-consistent framework. This enables the model not only to make predictions aligned with known physical laws but also to propagate gradients through those laws, allowing end-to-end training.

As a representative case study, we demonstrate this approach by combining CLOUD with the Debye model to predict temperature-

dependent phonon properties such as phonon internal energy ($U$) and heat capacity ($C_v$). They arise from collective lattice vibrations that are governed by the global periodicity of the crystal and the vibrational modes spanning multiple unit cells. According to the Debye model[56], $U$ and $C_v$ can be expressed as:

$$U \approx \frac{9}{8}Nk_B\Theta + 9Nk_BT\left(\frac{T}{\Theta}\right)^3\int_0^{x_D}\frac{x^3}{e^x-1}dx \qquad (7)$$

$$C_v = \left(\frac{\partial U}{\partial T}\right)_V \approx 9Nk_B\left(\frac{T}{\Theta}\right)^3\int_0^{x_D}\frac{x^4e^x}{(e^x-1)^2}dx \qquad (8)$$

$$x_D = \frac{\Theta}{T} \qquad (9)$$

$$\Theta = \frac{\hbar\omega_D}{k_B} \qquad (10)$$

where $\omega_D$ is the Debye frequency, $N$ is the number of atoms, $\hbar$ is the reduced Planck constant, $k_B$ is the Boltzmann constant, and $T$ is the temperature. The Debye frequency could be further written as $\omega_D = \left(\frac{6\pi^2N}{V}\right)^{\frac{1}{3}}v$ where $V$ is the volume of the $N$ atoms and $v$ is the speed of sound in the material. The sound velocity $v$ can be approximated using the first-order Hooke's law as:

$$v \approx \sqrt{\frac{C}{m}}d \qquad (11)$$

Here, $C$ denotes the effective spring constant, $m$ is the atomic mass within the primitive cell, and $d$ is the effective interplanar spacing along the direction of vibration. Therefore, $U$ and $C_v$ depend on lattice volume, bonding strength, and sound velocity – all of which are inherently long-range characteristics of the crystal structure. As a result, accurate prediction of $U$ and $C_v$ requires models capable of capturing these long-range structural dependencies and periodic constraints. However, recent systematic analyzes have shown that convolution-based Graph Neural Networks (GNNs) face inherent limitations in this regard[28,78]. Specifically, GNNs are prone to over-smoothing[79,80] and over-squashing[81,82], which restrict their receptive fields and impair their ability to learn long-range orders across the crystal. As a result, they often struggle with modeling properties that are sensitive to global structure. Transformer-based language models offer a promising alternative. Unlike GNNs, they capture interactions between all tokens without relying on sequential message passing, thereby enabling the modeling of long-range orders. Our proposed model, CLOUD, is built upon such a language model architecture and leverages the SCOPE representation, which directly encodes the space group–a global symmetry descriptor of the crystal. This makes CLOUD particularly well-suited for capturing the periodic and long-range features required for predicting phonon properties. To evaluate this capability, we evaluate CLOUD on the internal energy ($U$) and heat capacity ($C_v$) datasets curated by Gong et al[28]. These datasets, constructed from Materials Project entries[72], include DFT-computed phonon properties derived from phonon density of states (DOS) at 300K. By comparing performance on these tasks, we aim to assess whether language models can surpass state-of-the-art GNNs in learning the global structural representations necessary for accurate phonon property prediction.

Meanwhile, the original benchmarking experiments in the literature[28] are conducted by predicting the $U$ and $C_v$ values at $T = 300K$ directly from the models. Such an experimental setting comes with an obvious limitation that the model cannot learn the temperature dependency of $C_v$ and $U$. In other words, learning the mapping

**Table 1 | Results on predicting heat capacity ($C_v$) and phonon internal energy ($U$)**

| Model | $C_v$ (↓) | $U$ (↓) |
|---|---|---|
| CGCNN | 0.76 (0.03) | 0.71 (0.03) |
| ALIGNN | 0.63 (0.03) | 0.53 (0.03) |
| MEGNet | 0.71 (0.03) | 0.64 (0.02) |
| Matformer | 0.44 (0.02) | 0.46 (0.02) |
| de-CGCNN | <u>0.058 (0.004)</u> | 0.060 (0.005) |
| de-ALIGNN | 0.089 (0.006) | 0.10 (0.01) |
| de-MEGNet | 0.058 (0.005) | <u>0.057 (0.005)</u> |
| de-Matformer | 0.13 (0.01) | 0.14 (0.01) |
| CLOUD | 0.16 (0.00) | 0.15 (0.00) |
| CLOUD-DEBYE$_{scratch}$ | 0.091 (0.002) | 0.11 (0.01) |
| CLOUD-DEBYE$_{FT}$ | **0.057 (0.001)** | **0.055 (0.003)** |

All results are shown in the mean of test MAE/MAD in five-fold cross-validation, along with the standard deviation in brackets. CLOUD-DEBYE is evaluated for both trained-from-scratch (denoted as CLOUD-DEBYE$_{scratch}$) and pretrained-then-finetuned (denoted as CLOUD-DEBYE$_{FT}$). Descriptor-hybridized GNNs are denoted as de-GNN.
The best results are in bold, and the second-best results are underlined.

between crystal structures and those properties is intrinsically ill-posed. Possible solutions include training different models for different temperatures or encoding temperature explicitly in the crystal embeddings, but both of them are computationally expensive as $C_v$ labels under different temperatures are required, and they do not satisfy the thermodynamic consistency. According to Debye model, the phonon internal energy and heat capacity per unit cell is given by Equations (7) and (8), respectively[56]. Contrary to $C_v$ and $U$, the Debye temperature $\Theta$ is not dependent on the temperature but rather an intrinsic property of crystal structures. Gong et al.[28] claimed that the concatenation of descriptors improves the modeling of $\Theta$, whereas the thermodynamic consistency is still not achieved. Therefore, we propose CLOUD-DEBYE which is an integration of CLOUD and Debye model – CLOUD predicts $\Theta$ which is fed to Debye model for predicting $C_v$ and $U$. The differentiable implementation of Debye model enables training with $C_v$ or $U$ labels end-to-end without requiring the knowledge of Debye temperature.

The evaluation results for predicting $U$ and $C_v$ are presented in Table 1. The model performance is evaluated according to the metric MAE/MAD. Usually, a model will be considered as a good predictive model if MAE/MAD < 0.2[21,83]. While all the GNNs result in high MAE/MAD values, CLOUD, without the integration of Debye model, reaches 0.16 and 0.15 on $C_v$ and $U$, respectively. CLOUD-DEBYE further reduces the MAE/MAD metric significantly, outperforming all the descriptor-hybridized GNNs proposed by Gong et al.[28] which aimed to compensate for the lack of long-range information by concatenating global descriptors. The results manifest CLOUD's capability of learning long-range interactions in crystal structures as well as the significance of introducing thermodynamic consistency into the material machine learning framework. Interestingly, CLOUD-DEBYE achieves improved prediction accuracy than the purely data-driven CLOUD model pre-trained on ~ 6.3M data, even when trained from scratch. This underscores the importance of correctly incorporating physical principles: by enforcing thermodynamic consistency through the Debye model, CLOUD-DEBYE benefits from a strong inductive bias that leads to accurate and physically valid predictions. These results highlight that proper integration of physics is as important as, if not more important than, advances in machine learning techniques for building scientific foundation models.

We further visualize the error in $C_v$ prediction ($C_{v,\text{target}} - C_{v,\text{predict}}$) by different models in Fig. 4b. Compared to CGCNN[19] and ALIGNN[21], two competitive GNNs for material property

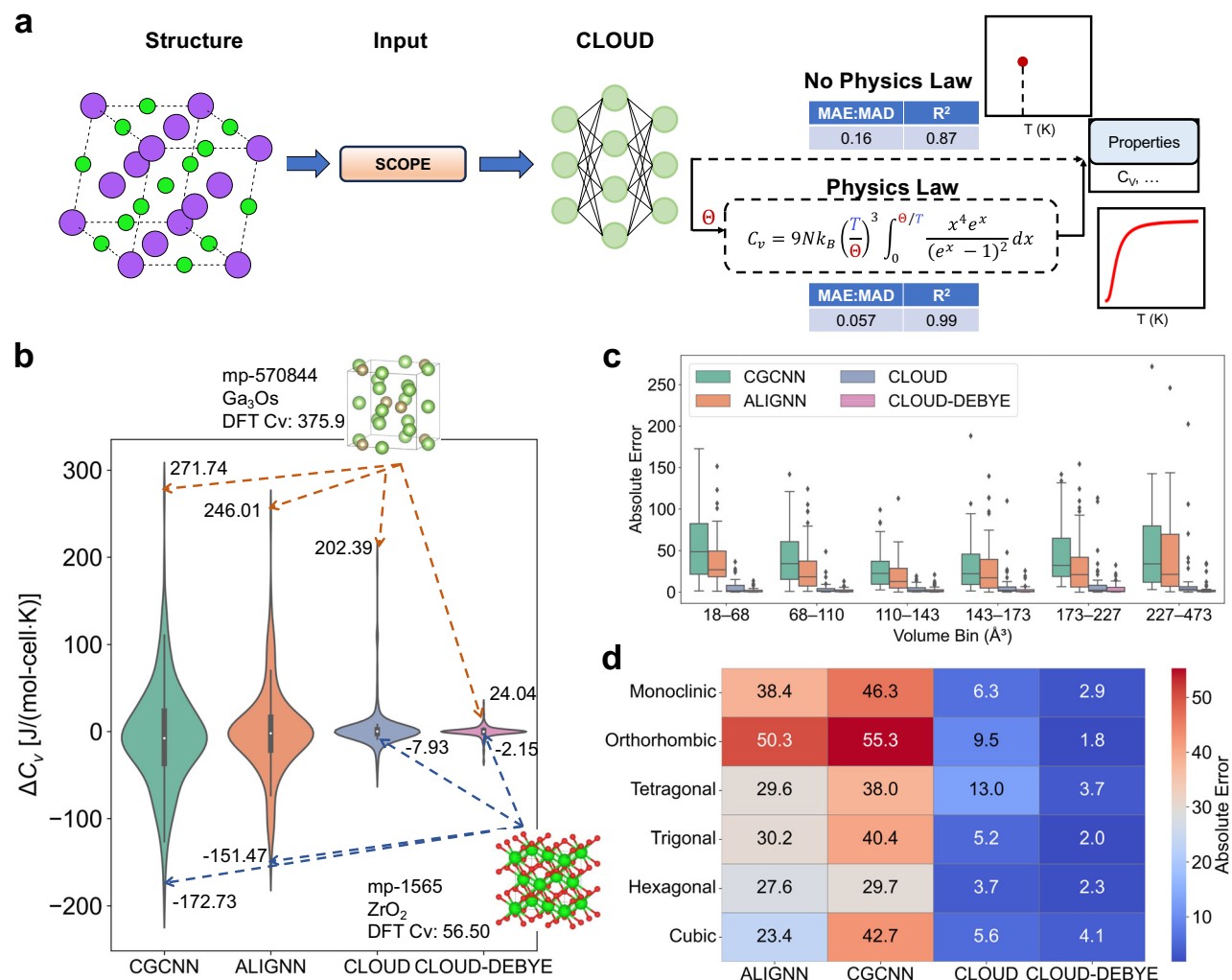

**Fig. 4 | CLOUD-DEBYE, a differentiable-physics framework with CLOUD.**
**a** Integration of physics laws with CLOUD model. The example exhibits the prediction for constant-volume heat capacity ($C_v$), where CLOUD outputs the Debye temperature ($\Theta$) and the Debye model predicts $C_v$ using $\Theta$ as input under different temperatures. **b** The violin plot of the prediction errors in $C_v$ by CGCNN, ALIGNN, CLOUD, and CLOUD-DEBYE. Two example structures from the $C_v$ test set are shown: mp-570844 (Ga₃Os) and mp-1565 (ZrO₂), and the corresponding residual $\Delta C_v$ and $C_v$ targets from the test set are marked in the figure. The crystal structures are visualized with VESTA[107]. **c** Box plot of absolute errors in $C_v$ prediction across six equal-sized bins sorted by primitive cell volume. Each bin contains approximately the same number of data points to ensure a balanced comparison. The center line denotes the median, box limits indicate the upper and lower quartiles, whiskers extend to 1.5 × the interquartile range, and points represent outliers. **d** Heatmap of $C_v$ prediction errors grouped by crystal system. Each cell shows the mean absolute error for a given space group, with crystal systems annotated along the y-axis for symmetry context.

predictions, which exhibit wide distributions and high variability of $\Delta C_v$, the CLOUD and CLOUD-DEBYE error distributions exhibit much smaller spread. Noticeably, CLOUD-DEBYE results in a much narrower error distribution and substantially reduces the maximum absolute error by more than a factor of six, suggesting that integration with the Debye model leads to more consistent $C_v$ predictions. Two representative structures, Ga₃Os (mp-570844) and ZrO₂ (mp-1565), are highlighted to illustrate this effect: while CGCNN and ALIGNN exhibit errors exceeding 150–270 J/(mol·cell*K), CLOUD-DEBYE produces much more accurate predictions, reducing the error to 6.5% of the target value from the dataset, or even lower. Furthermore, CLOUD-only fails to lower the error and still results in unphysical predictions without the integration of the Debye model. Such a result indicates that even when using the same model architecture and training objective, physics integration constrains unphysical outliers and enforces more reliable predictions. Figure 4c, d further examines the error distribution with respect to structural characteristics. In Fig. 4c, the test set data are grouped by primitive

cell volume, revealing that CGCNN[19] and ALIGNN[21] struggle particularly with large-volume structures, often producing substantial or unphysical errors. This trend reflects the limitations of GNNs with finite receptive fields, which hinder their ability to capture long-range interactions essential for accurate $C_v$ prediction in extended systems, consistent with the observation by Gong et al.[28]. In contrast, CLOUD significantly reduces the error across all volume bins, and CLOUD-DEBYE further improves upon this, achieving low and stable errors even for the largest structures. Figure 4d presents a heatmap of average error across crystal systems, where CLOUD-DEBYE consistently outperforms all GNNs and purely data-driven CLOUD. The error reduction is especially notable for complex systems such as orthorhombic, tetragonal, trigonal, and hexagonal crystals, underscoring the generalizability of physics-informed modeling across diverse symmetry classes. Therefore, the results highlight CLOUD's superior accuracy and robustness for learning features relevant to $C_v$ prediction, and also demonstrate the potential of combining it with physics-based models for real-world material applications.

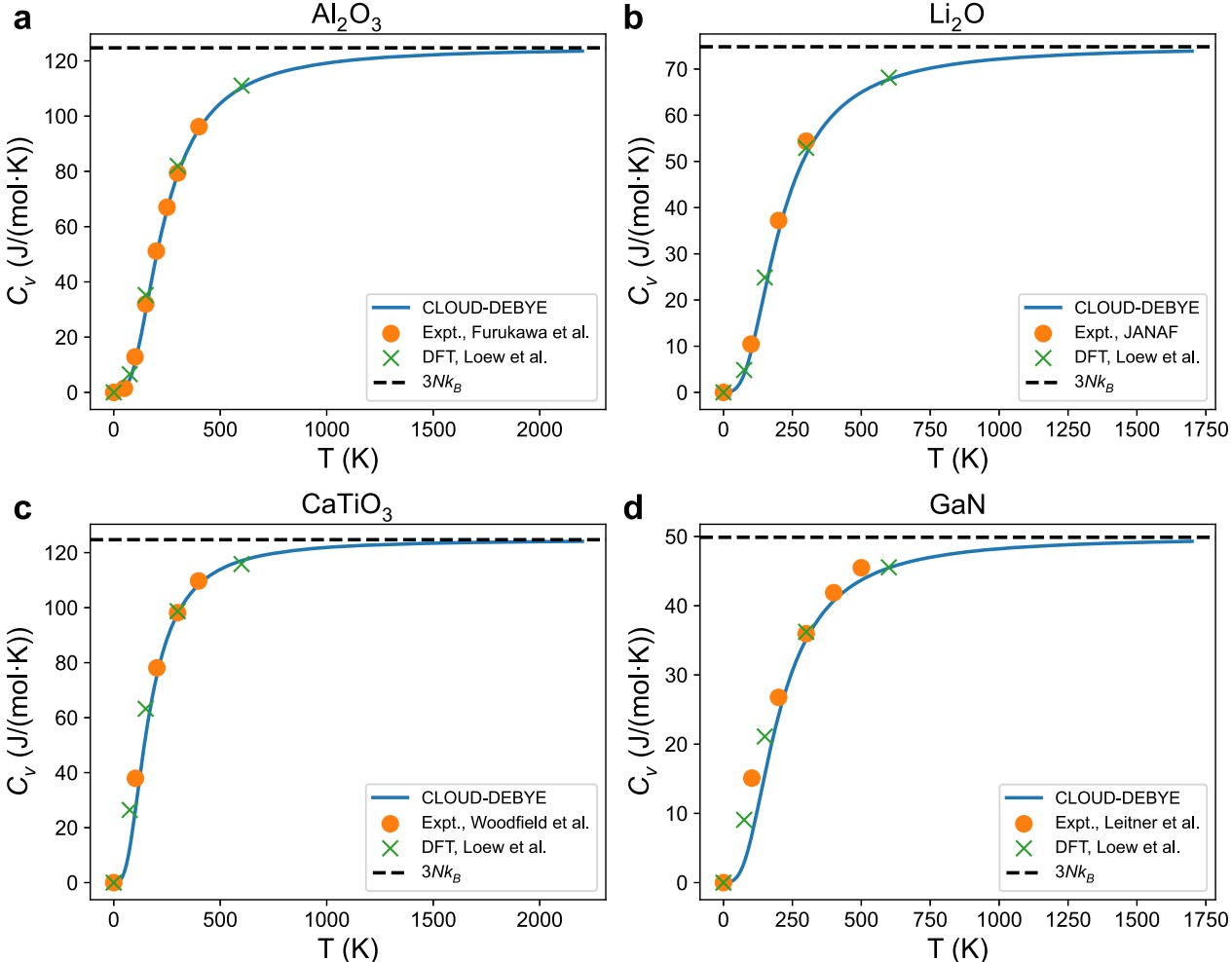

**Fig. 5 | Temperature-dependent heat capacity predicted by CLOUD-DEBYE.** The model predicts the constant-volume heat capacity ($C_v$) of crystalline materials across a wide temperature range: **a** $Al_2O_3$, **b** $Li_2O$, **c** $CaTiO_3$, and **d** GaN. The CLOUD-DEBYE model, fine-tuned on $C_v$ data at 300K, accurately extrapolates heat capacity across a wide temperature range, approaching the melting point of each material. Model predictions are compared against experimental[84–87] and DFT-calculated results[84]. The Dulong-Petit limit ($3Nk_B$) is included as a reference.

After being trained on $C_v$ data at $T = 300K$, CLOUD-DEBYE can be extrapolated to predict $C_v$ at other temperatures. We evaluate the model performance on different chemical systems: $Al2O_3$ and $Li_2O$ are two oxides, $CaTiO_3$ belongs to perovskite, and GaN is a III-V compound which is broadly used as a semiconductor. We vary the temperature from 0 K to near the melting points of the compounds. The predictions are compared to experimental results reported in previous literature[84–87] as well as phonon calculation results with Perdew-Burke-Ernzerhof (PBE) as the exchange-correlation functional from the Materials Data Repository (MDR) database[88]. As experiments are usually conducted under constant pressure, the experimental results are reported as constant-pressure heat capacity ($C_p$) which differs from $C_v$ by $C_p - C_v = \frac{\alpha^2 TV}{\kappa_T}$ where $\alpha$ is the expansion coefficient and $\kappa_T$ is the isothermal compressibility. However, $C_p$ is usually a good approximation for $C_v$ of solids under low temperature[56].

The prediction results by CLOUD-DEBYE are visualized in Fig. 5. For all the four systems that are experimented with, the CLOUD-DEBYE's prediction results match the experimental and DFT-calculated values well under different temperatures, even though the model is only trained with $C_v$ under 300K. In addition, $CaTiO_3$ is not included in either of the training or test set, which reflects the superior generalization performance of our model. Furthermore, the integration of

Debye model ensures the thermodynamic consistency of model predictions. According to Debye model, the heat capacity approaches

$$C_v \to 9Nk_B \left(\frac{T}{\Theta}\right)^3 \int_0^{x_D} x^2 dx = 3Nk_B \qquad (12)$$

as $T \to \infty$, which is expected (Dulong-Petit Law), since at high temperatures all phonon modes become activated. In the limit of $T \to 0$,

$$C_v \to 9Nk_B \left(\frac{T}{\Theta}\right)^3 \int_0^\infty \frac{x^4 e^x}{(e^x - 1)^2} dx \sim Nk_B \frac{T^3}{\Theta^3} \qquad (13)$$

Both of the two constraints are satisfied by the model predictions.

The results above strongly suggest the advantage of combining deep learning model with physics-based model as the temperature dependence of $C_v$ is directly embedded in the model, improving the data efficiency along the temperature dimension. Previous literature[11] has discussed the challenges of developing empirical[89,90] or machine-learning models for temperature-dependent heat capacity, primarily due to the difficulty of obtaining thermochemical property data as a function of temperature. Gurunathan et al.[50] alleviated this problem with a DOS-mediated approach: the ALIGNN force field[21] is trained to predict phonon Density of States (pDOS) from which the phonon-

based properties including $C_v$ could be derived. In contrast, our model integrates Debye model into the learning framework so that it can be trained end-to-end with $C_v$ data, learning the features to predict $\Theta$ from the crystal structures while not requiring any DFT-calculated labels of pDOS or $\Theta$. We expect the model to serve as a differentiable framework that learns essential physical properties of crystal structures from high-quality and easy-accessible data and predicts material properties while observing physics consistency.

Looking ahead, the CLOUD-DEBYE framework exemplifies a broader approach: embedding differentiable physics models into machine learning for physics-consistent predictions. Beyond the heat capacity and phonon internal energy, the same principle could be applied to other crystal properties governed by well-established physics laws. Examples include learning the full phonon density of states to access multiple vibrational observables, deriving elastic moduli through Voigt-Reuss-Hill averaging of elastic tensors, and modeling thermal expansion via the Grüneisen parameter and bulk modulus. We view these as promising directions for extending CLOUD into a general framework for physics-informed property prediction, and we provide further discussion in the section S3.5 in the Supplementary Information.

## Discussion

In this work, we introduce CLOUD, a scalable and physics-informed foundation model for crystal representation learning that combines symmetry-consistent representations with large-scale pretraining and finetuning. CLOUD leverages a symmetry-consistent string representation, SCOPE, to encode space-group symmetries, Wyckoff positions, and elemental compositions in a compact, coordinate-free format. Pre-trained on over six million unlabeled crystal structures and fine-tuned on labeled data of diverse downstream tasks, CLOUD achieves competitive performance across a wide range of material property prediction tasks, while exhibiting strong generalization in data-limited and out-of-distribution scenarios.

Our findings reveal several key takeaways. Firstly, the explicit encoding of symmetry significantly improves the predictive accuracy over previous coordinate-free approaches or even GNNs that directly encode 3D atomic coordinates. SCOPE forms a hierarchical representation consisting of global symmetries to equivalent sites and material compositions, thus accomplishing an efficient representation of crystal structures. The strong attention scores toward space group tokens in the SCOPE representation further confirm the model's capability to prioritize physically meaningful structural descriptors. The results highlight the significance of integrating symmetry constraints directly into the material representations, which receives little attention in most of the prior models that treat crystal structures as generic graphs or sequences.

In addition, we demonstrate that scaling laws observed in natural language processing models also apply to the foundation model for crystals. CLOUD exhibits strong scaling across both model and dataset sizes, closely aligning with Hoffmann's scaling framework. This insight opens the door to building larger, more powerful crystal foundation models that scale predictably with resources.

Moreover, CLOUD can learn long-range structural information. Traditional GNN-based methods often suffer from limited receptive fields due to over-smoothing and over-squashing. In contrast, the transformer architecture of CLOUD, combined with symmetry-consistent representations, enables superior performance on tasks that depend on global structural features, such as predicting heat capacity and internal energy. This is particularly evident in the integration with the Debye model, where CLOUD-DEBYE not only achieves better predictive performance but also ensures thermodynamic consistency across wide temperature ranges.

The integration of physics laws into differentiable frameworks offers a compelling direction for scientific machine learning, enabling models to produce predictions that are not only accurate but also physically consistent. By embedding analytical expressions, such as the Debye model, within the learning pipeline, CLOUD learns to predict physically meaningful intermediate quantities like the Debye temperature and propagates gradients through the physical laws to enforce thermodynamic constraints. This end-to-end differentiable setup contrasts with prior approaches that either predict target properties directly or rely on post hoc corrections. Our results show that incorporating physical laws into foundation models is not merely an enhancement but a critical design principle for achieving robust, generalizable, and scientifically sound predictions.

Several promising directions remain for future exploration. Despite CLOUD's strong scaling performance with respect to data size, the availability of high-quality crystal data for pretraining remains limited. Therefore, constructing a large-scale, high-quality crystal structure database would be instrumental in advancing foundational scientific models of crystals. Moreover, the current design of SCOPE can be further refined. Currently, SCOPE represents an ensemble of crystal structures that share the same space group, Wyckoff positions, and composition but differ in atomic coordinates due to the presence of free parameters in many Wyckoff positions. Additionally, we note two current limitations of the SCOPE representation. First, since Wyckoff labels depend on the input atom order, the representation is not strictly permutation-invariant, though this can be addressed by adopting a canonical ordering of Wyckoff sites. Second, SCOPE cannot fully capture partial occupancies or site mixing, as it encodes only the combined occupancy rather than element-specific assignments. Both aspects highlight promising directions for future work. We have provided a more detailed and in-depth discussion on limitations and future work in S1 Limitations and Future Work in the Supplementary Information.

In a nutshell, CLOUD serves as a unified foundation model that bridges machine learning and domain physics, providing a scalable and physically consistent framework for crystal property prediction. Its success in generalization, efficiency, and integration with physics laws makes it a promising tool for accelerating discovery in materials science and beyond.

## Methods

### Symmetry-consistent ordered parameter encoding

We design SCOPE, a symmetry-consistent string representation for crystal structures that integrates symmetry, equivalent sites, and constituting atoms to efficiently encode the structural and compositional information, eliminating the need for coordinate information or equivariant models. A SCOPE string for a crystal structure starts with the generator string which represents the basic symmetry operators of the corresponding space group. To build the generator string for each space group, all non-translational symmetry operations are represented by 14 generator matrices labeled a to n, while 10 uppercase letters encode translation components. Together, these generators define all symmetry operators and enable encoding all 230 space groups in a compact 4104-byte ASCII file[65].

Take the No. 35 space group as an example. Its generator string is given by '03aDDDbOOOjOOO0'. The '0' at the beginning of the generator string indicates that the inversion operator is not a generator. After that, '3' indicates that there are three generator matrices. Every four letters correspond to a generator: the first lower-case letter determines which of the 14 matrices is to be used. and the subsequent 3 letters stand for the translation components in three dimensions. For instance, 'aDDO' corresponds to the symmetry operator:

$$g = \begin{pmatrix} \mathbf{O} & \mathbf{t} \\ \mathbf{0} & 1 \end{pmatrix} = \begin{pmatrix} 1 & 0 & 0 & \frac{1}{2} \\ 0 & 1 & 0 & \frac{1}{2} \\ 0 & 0 & 1 & 0 \\ 0 & 0 & 0 & 1 \end{pmatrix} \tag{14}$$

The last symbol in the generator string is '0', which indicates that there is no alternative choice of the origin for this space group.

We argue that generator strings are more informative than space group symbols, as space group symbols only provide a high-level classification of symmetry, whereas generator strings offer detailed insight into the specific symmetry operations that define the crystal's structure. For example, $Cmc2_1$ (No. 36) and $Amm2$ (No. 38) only have one character in common in the space group symbols. However, their generator strings are '03aDDObOODjOOD0' and '03aODDbOOOjOOO0', respectively, suggesting that their symmetry generators are quite similar and only differ in the translation components. In fact, both space groups belong to the orthorhombic crystal system and exhibit mirror planes and two-fold rotation axes, but the subtle difference in the translation components (specifically in glide operations) reflects the underlying structural variation. This distinction is not apparent from the space group symbols alone, highlighting the value of generator strings in capturing more nuanced symmetry information. In contrast, $Fmm2$ (No. 42) has a face-centered lattice while $Imm2$ (No. 44) has a body-centered lattice, and the difference in lattice type significantly affects the symmetry operations and translations within the unit cell, even though they both belong to the Orthorhombic crystal system. Their space group names have 75% of the characters in common. Instead, their generator strings, '04aODDaDODbOOOjOOO0' and '03aDDDbOOOjOOO0', reflect more difference in the symmetry operators than their names.

Wyckoff positions in a SCOPE string are denoted by a symbol containing a number and a letter, e.g., 4d. The number denotes how many elements are on the corresponding orbit, also known as the "multiplicity" of the site. The letter is used to distinguish the Wyckoff positions in a given space group. The assignment rule for letters is that the letters showing up earlier in the alphabet are given to sites with lower multiplicity.

Finally, the generator string for the space group, the Wyckoff symbols for atomic positions, and material compositions (e.g., NaCl, $Al_2O_3$, $CaTiO_3$, etc.) are integrated to form a coordinate-free representation of crystals while maintaining crucial structural information like symmetries, enabling efficient encoding of crystal structures so that large language models can be utilized for crystal representation learning. To construct the SCOPE representation, we parse CIF files using pymatgen[91] to extract structural and symmetry information. For each structure, we apply 'SpacegroupAnalyzer' to determine its space group number and Wyckoff positions, and consequently retrieve the generator strings corresponding to the space groups. Finally, the material composition is extracted from the structure's fractional composition and appended to form the complete string. This results in a sequence of the form: [generator string] | [Wyckoff symbols] | [composition], efficiently capturing structural symmetry and stoichiometry without relying on atomic coordinates.

## Datasets

Pre-train Dataset: for CLOUD pre-training, we collect ~6.3M unique crystal structures from Open Databases Integration for Materials Design (OPTIMADE)[55] API. OPTIMADE integrates data from various major materials databases and provides an application programming interface (API) to make the data accessible to users. The contributing databases include: AFLOW[92], Alexandria[93], Computational Two-Dimensional Materials Database (C2DB)[94], Crystallography Open Database (COD)[95], Joint Automated Repository for Various Integrated Simulations (JARVIS)[96], Materials Cloud[97], Materials Platform for Data Science (MPDS)[98], Materials Project[72], Material-Property-Descriptor Database (MPDD)[99], Material Properties Open Database (MPOD)[100], NOMAD[101], Open Database of Xtals (odbx)[102], Open Materials Database (omdb)[103], and Open Quantum Materials Database (OQMD)[104]. Therefore, such comprehensive datasets enable effective pre-training on crystal representations. The original data we downloaded from

OPTIMADE consists of ~13M CIF files. To remove duplicate entries, we apply the deduplication rule from Antunes et al.[44]: structures are first grouped by identical chemical composition and space group number, and within each group, we retain only the structure with the smallest volume per formula unit. This criterion follows the observation that structures with identical composition and space group but larger volumes are often redundant polymorphs or duplicates of less compact configurations. Applying this rule to our dataset removes ~51.5% of the total structures, resulting in ~6.3M structures for subsequent processing. Finally, we use the de-duplicated data as the pre-training set.

Fine-tune Datasets: We evaluate CLOUD across a range of materials datasets that test different aspects of model performance. These include:

- MatBench[70], a diverse suite of materials property prediction tasks based on DFT and experimental data. We focus on eight regression tasks that include structural information and are widely used to assess model accuracy across various physical and chemical properties.
- MatBench Discovery[71], which focuses on predicting the stability of novel materials using models trained on customized formation energy data. The evaluation set comprises approximately 257,000 out-of-distribution crystal structures generated through elemental substitution[73]. Summary results are provided in the main text, with detailed evaluation in the Supplementary Information.
- UnconvBench[75], a dataset of unconventional crystal systems, including low-dimensional materials, metal-organic frameworks (MOFs), and defected structures. It provides a challenging testbed for assessing model robustness beyond well-ordered bulk crystals. Key results are summarized in the main text, with full analysis in the Supplementary Information.
- Phonon thermodynamics dataset[28], which includes phonon internal energy ($U$) and constant-volume heat capacity ($C_v$) computed at 300K for Materials Project structures[72]. These properties are derived from the phonon density of states (DOS) and reflect long-range structural periodicity and bonding strength. We use this dataset to examine CLOUD's ability to capture global structural features relevant for thermodynamic property prediction.

We include the information about the fine-tuning datasets that are used in this work in Table 2. More information about the fine-tuning datasets can be found in section S2.1 in Supplementary Information.

## Training details

The CLOUD model is developed upon the transformer encoder architecture[34,35]. Specifically, it employs Scaled Dot-Product Attention, which computes attention scores by measuring query-key alignment, scaling by the dimension of key, and applying softmax to derive weighted values:

$$\text{Attention}(Q, K, V) = \text{Softmax}\left(\frac{QK^T}{\sqrt{d_k}}\right)V \qquad (15)$$

Multi-head attention extends single attention by applying multiple linear projections to Q, K, and V, enabling parallel attention computations. The resulting outputs are concatenated and reprojected to form the final representation, allowing the model to effectively capture diverse subspace information.

CLOUD is first pre-trained via MLM on the dataset collected from OPTIMADE. The model is pre-trained in a self-supervised learning approach: without requiring the explicit supervision of labeled data, the model leverages intrinsic structure or patterns within the input data for learning. During pre-training, 15% of the tokens in a sequence are selected for potential modification, among which 80% are masked,

**Table 2 | Summary of the 8 regression tasks from MatBench, MatBench Discovery datasets, the 6 general predictive tasks from UnconvBench, and two phonon-related datasets $C_v$ and $U$ from Gong et al.[28] used in this paper**

| Dataset | Task | Size | Property | Unit |
|---|---|---|---|---|
| MatBench | jdft2d | 636 | Exfoliation energy of 2D materials | meV/atom |
| | phonons | 1265 | Highest phonon peak frequency | cm$^{-1}$ |
| | dielectric | 4764 | Refractive index | unitless |
| | gvrh | 10987 | Log10 of VRH average shear moduli | log GPa |
| | kvrh | 10987 | Log10 of VRH average bulk moduli | log GPa |
| | perovskites | 18928 | Formation energy of perovskites | eV/unit cell |
| | band gap | 106113 | Band gap | eV |
| | e form | 132752 | Formation energy | eV/atom |
| MB Discovery | MP (train) | 154718 | Formation energy | ev/atom |
| | WBM (test) | 256963 | Formation energy | ev/atom |
| UnconvBench | 2d_e_exf | 4527 | Exfoliation energy of 2D materials | eV/atom |
| | 2d_e_tot | 3520 | Total energy of 2D crystals | eV |
| | 2d_gap | 3520 | Band gap of 2D materials | eV |
| | qmof | 5106 | Formation energy of MOFs | eV |
| | supercon | 1058 | Curie temperature | K |
| | defected | 530 | Formation energy of defects | eV/atom |
| Phonon[28] | $C_v$ | 1512 | Constant-volume heat capacity | J/(mol-cell*K) |
| | $U$ | 1512 | Phonon internal energy | kJ/mol-cell) |

10% are replaced with randomly chosen vocabulary tokens, and the remaining 10% are left unchanged. The strategy, which is consistent with the original implementation proposed in literature[35], ensures the model learns meaningful contextual embeddings while biasing representations toward the actual observed tokens. The pre-training dataset is split into a training and a validation set by 80/20. AdamW optimizer is used with a learning rate of $5 \times 10^{-5}$ along with a cosine learning decay. The model is trained with batch size 2048 for a total of 50 epochs.

For the downstream task fine-tuning, we add a randomly initialized MLP after the encoder to output the property predictions. For MatBench, UnconvBench, and $U$ and $C_v$ datasets, five-fold cross-validation is performed, and the model is evaluated according to the average score on the test set; for MatBench Discovery, the Materials Project data are split into the training and validation set by 95/5, and the model is evaluated according to the performance on the WBM test set. We perform hyperparameter tuning for each dataset separately for optimal model performance. We provide additional information on model implementation in section S2.2 in Supplementary Information. To ensure a fair comparison on datasets from MatBench, we limit the prediction head for the three pre-trained language models (MatInFormer[31], SLICES-BERT where we train on SLICES[30] with the same architecture as CLOUD, and our CLOUD in this work) to be a single-layer MLP to ensure consistency with MatInFormer architecture implementation[31]. All the baseline results except for SLICES-BERT are obtained from the MatBench leaderboards[70]. When fine-tuned on MatBench Discovery for crystal structure stability classification, the model is first trained to predict the formation energy of the given structure. Based on the predictions for formation energies, the energy above the convex hull ($E_{\text{hull}}$), the distance to the convex hull spanned by competing phases in the same chemical system, is readily obtained to determine the stability of the materials. A material will be classified as stable if its $E_{\text{hull}}$ is lower than a threshold. The stability threshold for $E_{\text{hull}}$, typically set to 0, could be dynamically adjusted in order to account for the systematic prediction shifts unique to each model. Therefore, we vary the threshold and take a record of the statistics such as true positive rate (TPR) and false positive rate (FPR), which are used for further analysis.

When fine-tuning the model on the phonon internal energy ($U$) and constant-volume heat capacity ($C_v$) datasets with the CLOUD-DEBYE framework, we integrate the pre-trained CLOUD with the Debye model. Specifically, CLOUD is trained to predict the Debye temperature ($\Theta$), a characteristic property of the crystal structure that governs the temperature dependence of $U(T)$ and $C_v(T)$ through the Debye model. This approach introduces an inductive bias grounded in solid-state physics, enabling the model to generalize more effectively across temperatures. Moreover, while experimental or high-fidelity computational labels for $C_v$ are relatively accessible, direct labels for $\Theta$ are more difficult to obtain. By leveraging the Debye model as an intermediate physical prior, CLOUD indirectly learns to predict $\Theta$ from the structure and derives $U(T)$ and $C_v(T)$ accordingly. To enable end-to-end training with $U$ or $C_v$ as supervision targets, we adopt a differentiable implementation of the Debye model (Equations (7) and (8)), where the integrals are evaluated using Gauss-Legendre quadrature in PyTorch to allow gradient backpropagation through the physical model.

## Scaling analysis

Transformer-based language models benefit from the implicit parallelism of the attention mechanism and exhibit predictable performance improvements with scale. Empirical studies have shown that model performance, often measured via cross-entropy loss, follows power-law scaling laws with respect to both the number of model parameters and the amount of training data[39]. A canonical form of the scaling law introduced by (author?)[39] is:

$$L(N, D) = \left[ \left( \frac{N_c}{N} \right)^{\frac{\alpha_N}{\alpha_D}} + \frac{D_c}{D} \right]^{\alpha_D} \tag{16}$$

where $N$ is the number of tokens, $D$ is the number of non-embedding model parameters, and $N_c, D_c, \alpha_N, \alpha_D$ are dataset-dependent empirical constants. This formulation captures the trade-off between model capacity and training data volume under a fixed compute budget.

To better capture the behavior of loss in realistic settings and account for the irreducible entropy of the data distribution,

(author?)[40] propose a refined scaling law:

$$L(N, D) = \frac{A}{N^{\alpha}} + \frac{B}{D^{\beta}} + E \qquad (17)$$

where $A$, $B$, $E$, $\alpha$, and $\beta$ are fitting parameters. The constant term $E$ reflects the intrinsic entropy of the data distribution and defines the lower bound on achievable loss.

We evaluate the scaling performance of the proposed CLOUD model by systematically varying the number of pretraining tokens and the number of non-embedding parameters in the BERT encoder. We fit the empirical scaling behavior using the Hoffmann scaling law Equation (17) to capture the interaction between model size and dataset scale.

The fitting process involves solving the following optimization problem:

$$\min_{A,B,E,\alpha,\beta} \sum_i \text{Huber}_\delta \left( \log \widehat{L}(N_i, D_i) - \log L_i \right) \qquad (18)$$

where $\widehat{L}(N_i, D_i)$ is the predicted loss from the scaling law and $L_i$ is the observed loss for the $(N_i, D_i)$ configuration. We use the Huber loss with threshold $\delta = 10^{-3}$, which provides robustness to outliers, as recommended in (author?) [40]. The optimization is performed using the L-BFGS algorithm[77], with multiple random initializations to avoid suboptimal local minima.

This analysis allows us to estimate the compute-optimal scaling trajectory for CLOUD. By identifying the exponents $\alpha$ and $\beta$, we can characterize how performance scales with additional compute and whether improvements are better obtained by increasing model size or dataset size. Similar to the findings of (author?)[40], we observe that balanced scaling of model size and data yields optimal performance under fixed compute constraints, underscoring the importance of data collection alongside model architecture design.

## Data availability

The datasets used in this study are publicly available in previously published sources. The original crystal structure and property data can be found in the corresponding literature[28,55,70,71,75]. The processed datasets generated in this study and used for model training are available in the project GitHub repository at https://github.com/BattModels/CLOUD. No restricted-access or privacy-protected data were generated or used in this study.

## Code availability

All code developed and used in this study is publicly available in the project GitHub repository at https://github.com/BattModels/CLOUD, and the archived version used in this study is available at Zenodo (DOI: 10.5281/zenodo.17666827)[105]. The repository includes scripts for data processing, model pre-training, fine-tuning, and evaluation.

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

## Acknowledgements

Computing resources were provided as part of the U.S. Department of Energy's (DOE) Innovative and Novel Computational Impact on Theory and Experiment (INCITE) Program (Award Number: INCITE2025; Awardee: V.V. and C.X.). This research used resources from the Argonne Leadership Computing Facility, a U.S. DOE Office of Science user facility at Argonne National Laboratory, which is supported by the Office of Science of the U.S. DOE under Contract No. DE-AC02-06CH11357 (Awardee: V.V. and C.X.). This work was supported by Los Alamos National Laboratory under the grant number AWD026741 at the University of Michigan (Awardee: V.V. and C.X.). C.X., S.Z., and V.V. thank Advanced Research Computing at the University of Michigan for providing computing resources.

## Author contributions

C.X., S.Z., and V.V. designed research; C.X., S.Z., and V.V. contributed to the conceptualization and methodology of the SCOPE representation and CLOUD/CLOUD-DEBYE framework; C.X. implemented the algorithms; C.X., S.Z., and V.V. analyzed data and wrote the paper; V.V. supervised the work; all authors modified and approved the manuscript.

## Competing interests

The authors declare no competing interests.
