## [Transparent Peer Review file · Nature Communications]

CLOUD: A Scalable and Physics-Informed Foundation Model for Crystal Representation Learning

Corresponding Author: Professor Venkatasubramaniam Viswanathan

Version 0:

Reviewer comments:

Reviewer #1

(Remarks to the Author)

In this work, Xu and coauthors reported a representation learning framework for crystal structures that encodes materials information from a symmetry aware text description of crystals without precise atomistic coordinates as input. I think this work is generally interesting, particularly because of its powerful ability to capture global information of crystals, and the competitive performance of the benchmark compared with other structure-partially-blind models. I would recommend publication of this work in Nature Communications, if my following comments and suggestions could be properly addressed.

1. I think the main merit of this kind of model that do not require precise coordinates is to facilitate materials discovery in terms of thermodynamics. In other words, people can use space group and composition to propose new structures, and employ this type of model to evaluate stability without explicitly calculating coordinates. In this sense, the authors should find a way to demonstrate the ability of this model to evaluate convex hull of structures, maybe doing some test in the materials discovery benchmark.
2. It would be great for the authors to analyze more about the intrinsic limitation of this model. One example I can think of is the inability to distinguish AB type compound that permute the occupancy of sites for the two elements.
3. It would also be great to analyze the robustness of this model to small perturbation of crystal structures, because such perturbation would change the symmetry classification of the crystal, which might hugely alter the text description. But such perturbation can exist in everyday research, from sources like set up of geometry optimizer, inaccuracy of MLFF, and even the tolerance of space group analyzer.
4. In addition to the Debye model, what are some other properties of crystal that can be modeled in this thermodynamic consistent and differentiable way? The authors should include some discussion about it.

(Remarks on code availability)

Looks good to me

Reviewer #2

(Remarks to the Author)

This work introduces a transformer-based framework for crystal representation learning. The representation is SCOPE, a coordinate-free crystal string that concatenates space-group generator symbols (rather than just space group numbers), occupied Wyckoff positions, and composition information. The model is pretrained on OPTIMADE structures and fine-tuned on a wide suite of downstream materials property benchmarks (MatBench, MatBench-Discovery, UnconvBench). The reported results show a broadly competitive performance compared to graph neural networks (GNNs) and better than composition-based models. Attention analysis suggests the model attends strongly to symmetry tokens, supporting interpretability claims. The authors also investigate the scaling-law fits related to the data/model size to performance, offering guidance for future scaling. The authors further present a CLOUD-DEBYE that predicts a Debye temperature and propagates it through a differentiable Debye model to obtain heat capacity and internal energy curves, improving accuracy

while enforcing thermodynamic consistency. Overall, the work is comprehensive and timely, covering symmetry-aware representations, foundation models, and physics-informed ML for materials. However, the work can be better organized to show better novelty and clarity. I suggest that authors have a revision before the work can be accepted.

1. One key innovation in this study is the replacement of space group numbers with string generator with symmetry operations. The concept seems appealing and makes sense intuitively, but an ablation study is missing. It is unclear how much gain comes from (a) generator-level symmetry tokens vs space group numbers, (b) inclusion of Wyckoff positions. Comparisons to MatInFormer are confounded by dataset size, tokenizer scope, and model capacity differences.
2. "Long-range order/interactions" claim is underspecified and under-validated. The manuscript attributes CLOUD's advantage on UnconvBench and phonon thermodynamics to capturing long-range effects but provides no operational definition or ablation test. Prior work shows that mainstream GNNs often fail to capture periodicity and long-range information. The statement will be much stronger if the impact of the string generator of space group numbers on disordered materials. It will be nice if the authors can conduct a controlled ablation study on (a) SG-number only, (b) SG Hermann–Mauguin tokens, (c) Full SCOPE w/o Wyckoff, (d) Symmetry-only (mask composition), (e) Composition-only for perfect and disordered crystals datasets.

Aside from the above two main points, there are a few minor points to be addressed as well

3. Distinguish "long-range order" (periodicity) from "long-range interactions" (physical couplings such as elastic or electrostatic); the manuscript alternates between them.
4. Clarify de-duplication rule ("smallest volume per formula + SG") and report how many items dropped.
5. Some relevant work are not cited. A few generative symmetry-aware representations e.g. WyCryst, WyFormer, and SymmCD—demonstrate the importance of explicit symmetry and Wyckoff encodings for both generation and property prediction.

(Remarks on code availability)

I had a look at the github repo. The code structure seems clean with a proper readme and yaml env files. I didn't run the code for reproduction but it seems straightforward.

Reviewer #3

(Remarks to the Author)

The authors present a framework for predicting the properties of crystalline materials and compare it with multiple deep learning models, which is a hot topic. However, I have some concerns about the novelty of the framework and the performance comparison. The lack of application of some models in specific material systems also means that there is a lack of proof of the effectiveness of the models in practical applications.

1. The authors perform deduplication on the OPTIMADE dataset, but more details are not shown.
2. The authors pre-trained on the OPTIMADE dataset, evaluated the performance on the MatBench dataset, and compared it with other deep learning models (CGCNN, ALIGNN). Have CGCNN and ALIGNN also been pre-trained? If not, the performance comparison is unfair.
3. Large-scale pre-training is very expensive. Have the authors considered the issue of computational time? training time, testing time, model parameters
4. PMCGNN's performance on the JARVIS-DFT dataset is already higher than ALIGNN, Matformer, etc. The authors should also compare. <https://pubs.acs.org/doi/10.1021/acs.jcim.4c01200>
5. The author also mentioned that the performance of the current model depends on a large amount of labeled data, but the author's pre-training-fine-tuning model also depends on downstream labeled data. The self-supervised model may better handle this problem: CDSSL, <https://doi.org/10.48550/arXiv.2408.17255>.

(Remarks on code availability)

Version 1:

Reviewer comments:

Reviewer #1

(Remarks to the Author)

The authors have addressed my comments properly, and I suggest acceptance of this paper to be published in Nature Communications.

(Remarks on code availability)

The code seems good to me

Reviewer #2

(Remarks to the Author)

The authors did a good job in addressing my comments. I don't have further questions

(Remarks on code availability)

code is in good working condition with proper instruction

Reviewer #3

(Remarks to the Author)

I thank the authors for including additional model comparisons and validation analyses, which help support the model's performance and are central to this study's motivation. That said, my earlier concern remains: the proposed CLOUD method does not consistently outperform PMCGNN across key metrics. According to Table 6, only Ehull (reported in eV) shows a strong result, with other measures lacking clear improvement. To strengthen the case for CLOUD, I encourage the authors to highlight its distinct advantages, whether in terms of scalability, robustness, or applicability in specific scenarios, to better demonstrate its unique value beyond the current benchmarks.

(Remarks on code availability)

I haven't run the code, but the documentation and instructions provided are very clear and I believe it is reproducible.

Response to Reviews on Article “CLOUD: A
Scalable and Physics-Informed Foundation Model
for Crystal Representation Learning”

Changwen Xu¹, Shang Zhu¹, Venkatasubramanian Viswanathan^{1,2*}

¹Department of Mechanical Engineering, University of Michigan.

²Department of Aerospace Engineering, University of Michigan.

*Corresponding author(s). E-mail(s): venkvis@umich.edu;
Contributing authors: changwex@umich.edu; shangzhu@umich.edu;

Manuscript ID: NCOMMS-25-45609

We are very grateful for the reviewers’ comments, suggestions, and valuable input on our manuscript. There is no doubt that these comments are valuable and very helpful for improving our manuscript substantially. In what follows, we would like to answer the questions you mentioned and give a detailed account of the changes made to the original manuscript. We strongly believe that the revised manuscript should meet the standards for publication in Nature Communications.

Reviewer #1 Comments

Comment: *In this work, Xu and coauthors reported a representation learning framework for crystal structures that encodes materials information from a symmetry aware text description of crystals without precise atomistic coordinates as input. I think this work is generally interesting, particularly because of its powerful ability to capture global information of crystals, and the competitive performance of the benchmark compared with other structure-partially-blind models. I would recommend publication of this work in Nature Communications, if my following comments and suggestions could be properly addressed.*

Author Reply: We sincerely thank the reviewer for the thoughtful and encouraging assessment of our work. We deeply appreciate the recognition of the novelty of our symmetry-aware representation, its ability to capture global crystal information, and its competitive benchmark performance. In the following, we will address each of the detailed comments and suggestions provided.

Comment 1: *I think the main merit of this kind of model that do not require precise coordinates is to facilitate materials discovery in terms of thermodynamics. In other words, people can use space group and composition to propose new structures, and employ this type of model to evaluate stability without explicitly calculating coordinates. In this sense, the authors should find a way to demonstrate the ability of this model to evaluate convex hull of structures, maybe doing some test in the materials discovery benchmark.*

Author Reply: We thank the reviewer for the comment. In terms of the materials discovery benchmark, we have included the benchmarking results on MatBench Discovery, which could be found in Figure 2b in the manuscript and Figure S1, Table S4, and Table S5 in the supplementary information. In summary, MatBench Discovery is designed to evaluate ML models for materials discovery, particularly for predicting the thermodynamic stability of materials. ML models are trained to predict formation energies of the material structure then compare them to the convex hull to determine the stability. Models are evaluated on WBM dataset which contains unrelaxed structures only and is out-of-distribution from the training set. Our model manifests competitive performance, with ROC-AUC score comparable to ALIGNN [1]. When a

threshold of 0 is used, our model shows comparable performance to Wrenformer [2], while we observe that CLOUD benefits from more negative thresholds which lead to higher precision and DAF metrics. Therefore, the empirical evidence suggests the potential of CLOUD for materials discovery only based on the structure template (space group and Wyckoff positions) and the composition of the material, reducing the needs for high-fidelity simulation for stability identification.

In addition to that, we have further examined the model’s ability for novel stable material discovery by directly simulating a prospective screening workflow on the WBM dataset. Specifically, we predict the energy above convex hull E_{hull} from the model’s formation energy predictions, enabling the ranking of candidate structures by their predicted thermodynamic stability. Using this ranking, we carry out material stability evaluations under practical budget constraints. We assess (i) Precision@k, which quantifies the fraction of true stable materials among the top-ranked candidates, (ii) the Enrichment Factor, which measures how strongly the model concentrates true stables in the top fraction of its ranking compared to random selection, and (iii) Cost-to- M -stable, which reports the number of candidates that must be evaluated before discovering M stable materials. These complementary metrics simulate how an experimentalist or high-throughput workflow would employ the model to prioritize follow-up calculations or experimental evaluations. We additionally pick one example of novel material that manifests thermodynamic stability from the WBM test data, comparing the model prediction with the existing convex hull to demonstrate the model’s capability in material discovery.

To demonstrate our model’s capability in material discovery, we have benchmarked on MatBench Discovery. The related analysis is included in the section Transferrable and scalable crystal representation learning with CLOUD in **Results**:

Aside from predicting the material properties from the relaxed crystal structures, we are also interested in assessing CLOUD’s capability for structure stability prediction without knowing the relaxed crystal structures from DFT calculations. Therefore, we fine-tune the pre-trained model on MatBench Discovery [2] which is designed to evaluate machine learning models for materials discovery, particularly for predicting the thermodynamic stability of materials. The dataset aims to reflect practical challenges in the discovery process by requiring predictions based on unrelaxed crystal structures, which avoid reliance on expensive DFT calculations. We fine-tune CLOUD on Materials Project [3] formation energy training data and test on WBM dataset [4] which contains $\sim 257\text{K}$ OOD crystal structures generated by systematically substituting elements in pre-existing structures from Materials Project. Model performance is evaluated in terms of the ROC-AUC score, which is

calculated by varying the stability threshold, recording the false positive rate (FPR, the fraction of unstable structures being misclassified as stable) and true positive rate (TPR, the fraction of stable structures that are correctly identified), plotting the ROC (Receiver operating characteristic) curve, and calculating the area under curve (AUC) score. In contrast to directly predicting the formation energy for relaxed structures where CLOUD gives a larger prediction error than many structure-based GNNs, CLOUD manifests comparable classification performance despite not using atomic coordinates, given that the model achieves the AUC score of 0.81 which is close to that of ALIGNN [1] and higher than that of CGCNN [5] and MEGNet [6], as shown in Figure 1. These results highlight CLOUD’s potential for accelerating materials discovery by effectively screening unrelaxed structures and directly predicting the stability of the relaxed ones, thereby minimizing reliance on costly DFT calculations.

Fig. 1: Receiver operating characteristic (ROC) curves for models evaluated on MatBench Discovery for structure stability classification. False positive rate (FPR) and true positive rate (TPR) are the fraction of nonstable/stable structures classified as stable. Area under curve (AUC) scores are calculated and presented in descending order.

More related discussion on MatBench Discovery results was included in the section S3.2 MatBench Discovery in the Supplementary Information:

We train the model on Materials Project [3] formation energy data from the v2022.10.28 MP release, then make predictions for the unrelated structures in the WBM dataset [4] which are generated via elemental substitution of MP source structures so that the generated structures are not included in the training set. When evaluating the models on

MatBench Discovery, we record the classification results under varied thresholds and plot the receiver operating characteristic (ROC) curve for CLOUD and the major structure-based models that are trained on MatBench. We plot the ROC curves with more models included in Figure S1 compared to Figure 2c. Though not comparable to machine-learned interatomic potentials (MLIPs) at this stage, CLOUD still shows solid performance compared with CGCNN, MEGNet, Wrenformer, etc.

We further examine the other metrics listed in Table S4. The regression metrics of CLOUD are again second to ALIGNN while outperforming CGCNN and MEGNet. The classification metrics for CLOUD are close to those of Wrenformer under the stability threshold of 0, as shown in Table S4.

We list the classification results by CLOUD under different stability thresholds in Table S5. Note that the true labels for the test data are derived with the threshold of 0, consistent with the benchmark setting [2], while the dynamic threshold applies to the model prediction. More negative thresholds will result in higher precision for CLOUD and subsequently higher DAF, which is also observed for models that are more optimistic in stability predictions like CHGNet [2]. However, the trade-off across metrics leads to decreased F1 score and TPR when a negative threshold is used.

To further demonstrate our model’s potential in material discovery workflows, we have added the following comment in the section Transferrable and scalable representation learning with CLOUD in **Results** at page 10 in the revised manuscript:

To further showcase the capability of CLOUD in material discovery, we conduct a prospective discovery-style evaluation. We use CLOUD’s predicted formation energies to derive E_{hull} , rank candidates by predicted stability, and quantify discovery efficiency using Precision@k, Enrichment Factor, and Cost-to- M -stable metrics. These metrics emulate a realistic screening workflow under finite resource constraints. As shown in Figure S2a–c, CLOUD substantially improves efficiency by reducing the number of DFT calculations required compared to random screening. In a case study on the Al–Fe system, CLOUD successfully identifies Al_2Fe as stable, despite its absence from the Materials Project convex hull, in agreement with results from the WBM dataset (Figure S2d). These results highlight the strong potential of CLOUD to accelerate the discovery of novel stable materials.

Meanwhile, We have added the following results and discussion in the S3.2 MatBench Discovery in the Supplementary Information:

In order to showcase the potential of CLOUD in material discovery tasks, we further carry out a prospective discovery-style evaluation to directly demonstrate CLOUD’s ability to screen for new stable materials. On the WBM dataset which contains material structures that are out-of-distribution (OOD) to the training data, we calculate the energy above hull (E_{hull}), so far the same as the evaluation for MatBench Discovery. We rank all test candidates by their predicted distance to hull, E_{hull}^{pred} , in ascending order. Fix the stability

threshold τ (e.g., 0, 0.02, or 0.05 eV/atom) and define binary labels

$$s_i = \mathbf{1}\left[E_{\text{hull},i}^{\text{true}} \leq \tau\right] \in \{0, 1\}, \quad (1)$$

for $i = 1, \dots, N$. Let π denote the permutation that sorts candidates by $E_{\text{hull}}^{\text{pred}}$ (best first), so $\pi(1)$ is rank-1, etc. We then quantify discovery efficiency under realistic screening budgets using three complementary metrics:

- **Precision@k**: the fraction of true stables in the top- k suggestions,

$$\text{P@}k = \frac{1}{k} \sum_{j=1}^k s_{\pi(j)}. \quad (2)$$

- **Enrichment Factor** at top $\alpha\%$ (with $k = \lfloor \alpha N \rfloor$): the fold improvement over random screening,

$$\text{EF}_\alpha = \frac{1}{k} \sum_{j=1}^k s_{\pi(j)} / \frac{1}{N} \sum_{i=1}^N s_i \quad (3)$$

- **Cost-to- M -stable**: the number of evaluations needed to uncover M true stables when following the ranked list,

$$\text{Cost-to-}M = \min \left\{ m : \sum_{j=1}^m s_{\pi(j)} \geq M \right\}. \quad (4)$$

For comparison to the theoretical optimum, we also report the cost factor $\text{Cost-to-}M/M \geq 1$, where 1 corresponds to the ideal case.

The workflow resembles the true material discovery process where the most promising candidates, rather than the whole pool, are sorted out for final evaluation with theoretical calculation or experiments. As shown in Figure 2a, the precision for stability screening remains high even up to the top-1000 candidates and reaches nearly 100% for $k \leq 200$, showing that the model consistently prioritizes true stable materials at the top of the ranked list. The enrichment factor further demonstrates that the model concentrates stable compounds effectively, with as much as ~ 10 -fold enrichment over random selection within the top fraction of candidates selected by the model (Figure 2b). The cost factor curves reveal that CLOUD identifies a target number of stable materials with only about 1.1–1.3 times the evaluations required by the theoretical optimal (Figure 2c), highlighting its efficiency under practical screening budgets.

Besides the metrics shown above, we pick one example from the WBM dataset, which is Al_2Fe (space group: No.139, I4/mmm), and compare it with the convex hull for the Al-Fe system obtained from Materials Project [3]. As shown in Figure 2, Al_2Fe , though not included in the Materials Project database, exhibits a lower formation energy than the existing convex hull. Our model CLOUD successfully identifies its stability, despite that the formation energy prediction itself is not perfectly accurate. Therefore, this case study demonstrates how CLOUD can highlight previously unseen stable materials from only symmetry-consistent templates and composition, with unrelaxed structures as input only, thereby reducing the number of expensive high-fidelity evaluations needed for discovery.

Fig. 2: Evaluation of CLOUD in materials discovery tasks on the WBM dataset. We vary the stability threshold $\tau \in \{-0.05, -0.02, 0.0, 0.02, 0.05\}$ and calculate the following metrics: **a** Precision@k: fraction of true stables recovered within the top-k predictions across stability thresholds; **b** Enrichment Factor: fold enrichment over random screening in the top $\alpha\%$ of the ranked list; **c** Cost-to- M -stable: number of evaluations required to uncover M stable materials, normalized by the ideal case (dashed line). **d** Case study: Al-Fe convex hull. CLOUD correctly identifies the intermetallic Al_2Fe as a new stable candidate.

Comment 2: *It would be great for the authors to analyze more about the intrinsic limitation of this model. One example I can think of is the inability to distinguish AB type compound that permute the occupancy of sites for the two elements.*

Author Reply: We thank the reviewer for this thoughtful comment. We agree that discussing the intrinsic limitations of our model is important for a comprehensive understanding of its scope and potential extensions.

One limitation arises from the ordering of atoms in the input structure. Although our SCOPE representation is theoretically invariant to symmetry operations, the specific assignment of elements to Wyckoff positions is determined by the atom order provided in the structure file. Since language models are inherently sensitive to token order due to positional embeddings, the current SCOPE encoding is not strictly invariant to permutations of the atomic list. To address this limitation, a natural remedy is to impose a canonical ordering of Wyckoff positions (e.g., sorting by site symmetry, starting from 1a, and permuting the elements accordingly), thereby removing dependence on input atom order.

A second limitation concerns partial occupancies. In SCOPE, each Wyckoff position is explicitly associated with its occupying element, which ensures that compounds with different element–Wyckoff assignments (e.g., AB-type structures occupying distinct Wyckoff sites) are distinguishable. However, when multiple elements partially occupy the same Wyckoff position, SCOPE does not encode which specific symmetry-equivalent sites are occupied by which element. Instead, only the combined occupancy is represented. While such cases are relatively rare in the benchmark datasets used in this work, this limitation becomes more relevant in modeling disorder and site-mixing phenomena. We view the extension of SCOPE to explicitly handle partial occupancies as an interesting direction for future work.

Finally, we also provide a broader discussion of limitations and future work in Section S1 of the Supplementary Information. These include the finite size of available pretraining data and the absence of certain structural details in SCOPE (such as free positional parameters within Wyckoff sites).

We have included the discussion on the limitation of shortage of pre-training data and missing some structural information in the section S1 Limitations and Future Work in the Supplementary Information:

While CLOUD shows competitive performance in crystal representation learning and property prediction, it does encounter limitations and potential improvements for future explorations. One limitation is that our model requires a large amount of pre-training data

to achieve SOTA performance in many of the downstream tasks according to the scaling law we fitted, whereas such a dataset has not yet been built. Currently, the largest crystal database we can build is the one collected from OPTIMADE which includes $\sim 6.3\text{M}$ unique crystal structures. Compared to molecular science in which ~ 50 billion unlabeled SMILES of molecules can be easily obtained [7–9], obtaining such a large number of crystal structures is challenging despite the large design space of crystals. Not all combinations of atoms can form valid crystal structures [10] as many potential configurations are inherently unstable or chemically incompatible. Recently, a lot of machine learning models have been proposed for crystal generation [11–14], however, none of them have pushed the data size to the order of billions. Recently, OMat24 dataset [15] was released which contains over 100 million DFT calculations for solid-state materials, which is a huge contribution to the research community to advance materials science development with AI. In future work, we plan to comprehensively integrate existing datasets like OMat24 and diverse generation approaches for crystal structure generation, ranging from rule-based methods to data-driven approaches. We will examine how the expanded, synthetic dataset impacts the model performance in both pre-training scaling and downstream predictive accuracy.

Another limitation lies in the representation itself: the representation we design is built on symmetry, whereas symmetry does not capture everything. For instance, while the space group, Wyckoff positions, and stoichiometry remain unchanged during relaxation, the atomic positions and cell parameters undergo significant variations. As a result, CLOUD is unable to leverage the MPtrj dataset, which comprises over 1.5 million crystal structures from relaxation trajectories, unlike many other models on the leaderboard [2, 16]. In addition, the representation does not work for amorphous materials which lack the long-range order and symmetry that crystalline materials exhibit, thus limiting the versatility of CLOUD in handling a broader range of materials. Furthermore, some Wyckoff positions include free variables, meaning the representation encodes an ensemble of materials. As a result, it may not distinguish between materials with different physico-chemical properties that share the same prototype but have different atomic positions. In future work, we plan to systematically investigate the design of string representations for crystals which enables effective encoding of more information besides symmetry and composition, while building the multi-modal framework for crystalline materials as an alternative to incorporate comprehensive structural information.

Plus, we have also mentioned the limitations and future work concisely in the **Discussion** section in the manuscript:

Several promising directions remain for future exploration. Despite CLOUD’s strong scaling performance with respect to data size, the availability of high-quality crystal data for pretraining remains limited. Therefore, constructing a large-scale, high-quality crystal structure database would be instrumental in advancing scientific foundation models for crystals. Moreover, the current design of SCOPE can be further refined. Currently, SCOPE

represents an ensemble of crystal structures that share the same space group, Wyckoff positions, and composition but differ in atomic coordinates due to the presence of free parameters in many Wyckoff positions.

In addition to the limitations we have already discussed in the original manuscript and the Supplementary Information, we have added content related to the limitations we have discussed in the response into the second-to-last paragraph of **Discussion** in the manuscript:

Additionally, we note two current limitations of the SCOPE representation. First, since Wyckoff labels depend on the input atom order, the representation is not strictly permutation-invariant, though this can be addressed by adopting a canonical ordering of Wyckoff sites. Second, SCOPE cannot fully capture partial occupancies or site mixing, as it encodes only the combined occupancy rather than element-specific assignments. Both aspects highlight promising directions for future work. We have provided a more detailed and in-depth discussion on limitations and future work in S1 Limitations and Future Work in the Supplementary Information.

Meanwhile, we have also added a detailed discussion into the second paragraph in the S1 Limitations and Future Work section in the Supplementary Information:

In addition, two further limitations deserve mention. First, although the SCOPE representation is invariant to symmetry operations by design, it is not invariant to permutations of the atom list in the input structure. Since Wyckoff labels are assigned in the order atoms are provided, permuting the atoms in a CIF can lead to different sequences, which is undesirable given that language models are sensitive to token order, and in future work we plan to solve it by enforcing a canonical ordering of Wyckoff sites (e.g., sorted by site symmetry, starting from 1a) and permuting elements accordingly. Second, while SCOPE distinguishes compounds with different element–Wyckoff assignments, it cannot fully resolve cases where multiple elements partially occupy the same Wyckoff position. In such situations, only the combined occupancy is encoded, rather than the specific site assignments of each element. This limitation becomes important for modeling disorder and site mixing, and extending SCOPE in this direction represents a promising direction for future work.

Comment 3: *It would also be great to analyze the robustness of this model to small perturbation of crystal structures, because such perturbation would change the symmetry classification of the crystal, which might hugely alter the text description. But such perturbation can exist in everyday research, from sources like set up of geometry optimizer, inaccuracy of MLFF, and even the tolerance of space group analyzer.*

Author Reply: We thank the reviewer for this valuable suggestion. We agree that robustness to small perturbations—such as those arising from geometry optimization settings, inaccuracies in ML force fields, or the tolerance thresholds of symmetry analyzers—is an important practical consideration for any structure-based representation.

Interestingly, a recent study by Siron et al. [17] — which cites our workshop paper [18] — has already performed a systematic robustness analysis that includes the SCOPE representation as one of the benchmarked methods. In their evaluation, multiple structure identification and fingerprinting approaches (including SCOPE, SLICES, Pymatgen’s StructureMatcher, and their proposed BAWL hashing) were tested for matching perturbed crystal structures. Perturbations included Gaussian noise on fractional atomic coordinates and on lattice vectors, as well as symmetry operations and translations. Their results (Table 1 and Figure 3) show that CLOUD maintains moderate robustness to small perturbations in atomic positions and lattice strain, though methods explicitly designed for perturbation invariance (e.g., BAWL) exhibit higher tolerance at large noise levels.

Table 1: Comparison of different methods matching pairs of disordered structures, adapted from Siron et al. [17]. The success rate is reported on all the combinations of pairwise similarity for different chemical formulas.

BAWL	Short-BAWL	EqV2-sim	PDD	Pymatgen	SCOPE	SLICES
0.14 ± 0.18	0.30 ± 0.33	0.61 ± 0.38	0.56 ± 0.46	0.00 ± 0.01	0.46 ± 0.33	0.00 ± 0.00

This independent analysis provides evidence that our SCOPE representation is reasonably stable under minor structural distortions, even though such distortions may change the exact space group classification and thus the tokenized description. We will cite these results in the revised manuscript and add a short discussion in the Methods section to acknowledge current robustness performance and potential future work.

In addition to the existing analysis on the robustness on the SCOPE representation, we have also investigated in the robustness of model predictions with respect to different noise levels on atomic positions and lattice vectors. Similar to the analysis by Siron et al. [17], we applied Gaussian noise of different levels of perturbations to mimic atomic coordinates noise or the lattice strain, and report how the model performance varies with the standard deviation σ of the noise.

Fig. 3: Success rate of structure identification methods under different different perturbations, adapted from Siron et al. [17]. The left figure shows the performance of each method under Gaussian noise added to the atomic positions, while the right figure shows the performance under lattice strain from a Gaussian distribution applied independently to each lattice vector.

We have added the following results and discussions on representation and model robustness under small perturbation of crystal structures in the section S3.6 Robustness Analysis in the revised Supplementary Information:

In addition to the property prediction results, we investigate the robustness of building SCOPE representations and CLOUD model predictions under controlled perturbations of the input crystal structures. Such perturbations are practically unavoidable in realistic materials research, arising from geometry optimization tolerances, noise in machine-learned force fields, or the thresholds used by symmetry analyzers. Because the SCOPE representation relies on symmetry parsing, even small distortions can alter the assigned space group or Wyckoff positions. We therefore probe how predictive accuracy and parsing stability degrade under systematically applied noise to atomic coordinates and lattice vectors, providing a quantitative measure of the sensitivity of our models to structural perturbations.

A recent independent study by Siron et al. [17] provides an external benchmark of this issue. They systematically compared multiple structure identification and fingerprinting approaches—including the SCOPE representation for CLOUD, Pymatgen’s Structure-Matcher, SLICES, and their proposed BAWL hashing—for matching perturbed crystal structures. Perturbations included Gaussian noise on fractional atomic coordinates and on lattice vectors, as well as symmetry operations and translations. Their results show that SCOPE (and thus CLOUD) maintains moderate robustness to small perturbations, outperforming other representations such as SLICES, while specialized perturbation-invariant methods such as BAWL achieve higher tolerance at larger noise levels. This analysis establishes that SCOPE is reasonably stable under minor structural distortions.

Building on this prior evaluation, we additionally perform a dedicated robustness study of our model predictions by perturbing each test crystal with (i) fractional-coordinate noise, adding i.i.d. $\mathcal{N}(0, \sigma^2)$ to fractional positions, and (ii) lattice strain, drawing a random symmetric small-strain tensor ε with i.i.d. entries $\mathcal{N}(0, \sigma^2)$, symmetrizing $\varepsilon = \frac{1}{2}(E + E^\top)$; this perturbs both lattice lengths and angles (normal and shear). For every perturbed structure we regenerate SCOPE and evaluate fixed checkpoints (no retraining) over a log grid σ from 10^{-4} up to at most 10^0 . Alongside prediction error (Figure 4a-c) we report the symmetry-parse success rate (Figure 4d). On the gvrh dataset from MatBench [19], the MAE for unperturbed structures is 0.0873 ± 0.0028 . Fractional coordinate noise induces a monotonic increase—0.1004 at $\sigma = 3 \times 10^{-4}$ (+15%) and 0.1201 at 10^{-3} (+37.6%)—whereas lattice strain shows much less significant impact on model performance up to 10^{-2} (MAE ≈ 0.090 , $\leq +4\%$). Parsing succeeds for all small- σ cases (rate = 1.0 through 10^{-2}); degradation appears only under heavy stress (e.g., at $\sigma = 0.1$ the success rate drops to 0.774 for coordinate noise and 0.987 for strain, and to 0.075 / 0.399 at $\sigma = 0.3$). On the C_v dataset, CLOUD is markedly more sensitive to coordinate noise: 8.92 \rightarrow 11.00 at 10^{-4} (+23%), 19.71 at 3×10^{-4} (+121%), and 27.04 at 10^{-3} (+203%); strain grows more gently (12.29 at 10^{-3} , 13.66 at 10^{-2}). Symmetry parse success remains ≈ 0.997 up to 10^{-2} , indicating that our method of constructing SCOPE for crystal structures is robust to small perturbation levels. Meanwhile, we need to point out that the symmetry parsing success does not necessarily guarantee that the true symmetry for the unperturbed structure could be recovered. Adding the physics head in CLOUD-DEBYE both lowers baseline error (3.19 ± 0.04) and dampens sensitivity: under coordinate noise at $\sigma = 10^{-3}$ the MAE is 5.27 (versus 27.04 for CLOUD), and under strain at $\sigma = 10^{-2}$ it is 4.66 (versus 13.66). Overall, these results demonstrate that CLOUD is reasonably robust to modest structural perturbations, with lattice strain having limited effect and coordinate noise being more impactful, while the physics-informed CLOUD-DEBYE variant substantially mitigates sensitivity and enhances stability across perturbation regimes.

Commet 4: *In addition to the debye model, what are some other properties of crystal that can modeled in this thermodynamic consistent and differentiable way? The authors should include some discussion about it.*

Author Reply: We thank the reviewer for the comment. In addition to the Debye heat capacity, we have extended our CLOUD-Debye framework to predict the vibrational entropy S_{vib} . This quantity arises consistently from the Debye model and is a distinct thermodynamic observable beyond C_v .

Beyond the Debye model, the same framework can naturally be extended to other properties governed by well-established physics laws. In principle, our differentiable physics framework is generalizable as long as there exist enough training data and the physics model could be implemented in a differentiable way. In such a framework,

Fig. 4: Model performance robustness analysis. Gaussian noise with varying σ 's are added to the atomic positions or lattice strain applied to lattice vectors. We show the Δ MAE due the noise for **a** gvrh with CLOUD model; **b** Cv with CLOUD model; **c** Cv with CLOUD-DEBYE model. In addition, the symmetry parsing successful rate under different noises are presented in **d**.

the differentiable physics model provide constraints as well as guidance to enforce physics consistency, while the complex mapping will be learned by the machine learning model in a data-driven manner. We have added the discussion on future work of extending the framework to a wider range of physics models, investigating the possibility of integrating CLOUD with the (i) pDOS for more general and accurate predictions of phonon-related properties; (ii) predicting elastic modulus from elastic tensors; (iii) predicting thermal expansion coefficient through the Grüneisen parameter γ and temperature-dependent bulk modulus. These cases demonstrate the generality of our approach in coupling learned representations with differentiable physical laws.

We have added one concise paragraph at the end of the section CLOUD-DEBYE: Integration of CLOUD in a differentiable and physics-consistent materials thermodynamics framework in **Results**:

Looking ahead, the CLOUD–Debye framework exemplifies a broader approach: embedding differentiable physics models into machine learning for physics-consistent predictions. Beyond the heat capacity and phonon internal energy, the same principle could be applied to other crystal properties governed by well-established physics laws. Examples include learning the full phonon density of states to access multiple vibrational observables, deriving elastic moduli through Voigt–Reuss–Hill averaging of elastic tensors, and modeling thermal expansion via the Grüneisen parameter and bulk modulus. We view these as promising directions for extending CLOUD into a general framework for physics-informed property prediction, and we provide further discussion in the section S3.5 in the Supplementary Information.

This paragraph forms a brief outlook for extending the CLOUD-DEBYE framework we proposed.

Meanwhile, we have also added the following discussion in S3.5 Extension of CLOUD-DEBYE: Towards Broader Applications in the revised Supplementary Information to demonstrate the generalizability of our differentiable physics approach in materials modeling.

We have extended our CLOUD-DEBYE framework for predicting vibrational entropy (S_{vib}) to showcase its broad application in many other property prediction tasks. Table 2 summarizes the results for predicting S_{vib} . Similar to our implementations for predicting constant-volume heat capacity C_v and phonon internal energy U , the train and test data are under 300K, and we use MAE/MAD as the metric for comparison. CLOUD-DEBYE outperforms the two descriptor-hybridized GNNs (de-CGCNN and de-ALIGNN) and CLOUD, which is consistent with the observations for C_v and U .

Table 2: Results on predicting S_{vib} . All results are shown in the mean of test MAE/MAD along with the standard deviation in brackets. The best results are in bold and the second-best results are underlined.

Model	S_{vib} (\downarrow)
de-CGCNN	0.154 (0.003)
de-ALIGNN	0.195 (0.005)
CLOUD	0.212 (0.009)
CLOUD-DEBYE	0.121 (0.004)

In addition, we extrapolate the model fine-tuned on 300K data to a wide range of temperatures and compare the model predictions to experimental [20–23] as well as phonon calculation results [24]. Similar to the observations for C_v , our model manifests superior predictive accuracy from 0K up to near the melting point of the solid, maintaining physical consistency in model predictions (Figure 5).

Fig. 5: S_{vib} predictions by CLOUD-DEBYE for **a** Al_2O_3 , **b** Li_2O , **c** $CaTiO_3$, and **d** GaN . The CLOUD-DEBYE model, fine-tuned on S_{vib} data at 300K, accurately extrapolates heat capacity across a wide temperature range, approaching the melting point of each material. Model predictions are compared against experimental and DFT-calculated results.

Our experimental results present a compelling demonstration of the significance of integrating CLOUD with physical models for physics-consistent predictions. Moreover, our framework of embedding physics-based, differentiable laws into machine learning models is not limited to the Debye model. In principle, this approach can be generalized to any physical system, provided that the underlying physics can be implemented in a differentiable manner and sufficient data are available to learn the complex mappings that remain intractable in the physics model. In the following, we will briefly discuss the possible extensions of the CLOUD-physics framework which can serve as future work:

- **Phonon-related properties from pDOS** While the Debye model provides a quadratic approximation to the phonon spectrum and thus misses optical branches and fine spectral features, these can be captured by predicting the full phonon density of states (pDOS). In this setting, the ML model predicts the pDOS $g(\omega)$ directly from the crystal representation, which is then fed into analytic statistical-mechanics relations for vibrational thermodynamics. For example, the constant-volume heat capacity C_v is given by

$$C_v(T) = \int_0^\infty \hbar\omega g(\omega) \left[\frac{1}{(e^{\hbar\omega/k_B T} - 1)^2} e^{\hbar\omega/k_B T} \frac{\hbar\omega}{k_B T^2} \right] d\omega, \quad (5)$$

and the vibrational entropy by

$$S_{\text{vib}}(T) = k_B \int_0^\infty g(\omega) \left[\frac{\hbar\omega/k_B T}{e^{\hbar\omega/k_B T} - 1} - \ln(1 - e^{-\hbar\omega/k_B T}) \right] d\omega. \quad (6)$$

Therefore, the mapping from pDOS to thermodynamic observables is fully differentiable, allowing gradients to backpropagate through the integrals, and (ii) a single ML-predicted $g(\omega)$ provides access to multiple vibrational quantities (e.g. C_v , S_{vib} , free energy), enabling multitask training and richer physics constraints.

- **Elastic modulus:** Elastic properties can be described by closed-form averages of the elastic tensor C_{ij} . The Voigt bounds depend linearly on C :

$$K_V = \frac{1}{9} (C_{11} + C_{22} + C_{33} + 2(C_{12} + C_{23} + C_{13})) \quad (7)$$

$$G_V = \frac{1}{15} [(C_{11} + C_{22} + C_{33} - C_{12} - C_{23} - C_{13}) + 3(C_{44} + C_{55} + C_{66})] \quad (8)$$

The Reuss bounds depend on the compliance tensor $S = C^{-1}$:

$$K_R = \frac{9}{S_{11} + S_{22} + S_{33} + 2(S_{12} + S_{23} + S_{13})} \quad (9)$$

$$G_R = \frac{15}{4(S_{11} + S_{22} + S_{33}) - 4(S_{12} + S_{23} + S_{13}) + 3(S_{44} + S_{55} + S_{66})} \quad (10)$$

Therefore, the Voigt-Reuss-Hill averages are

$$K_{VRH} = \frac{1}{2}(K_V + K_R), \quad G_{VRH} = \frac{1}{2}(G_V + G_R) \quad (11)$$

All mappings are smooth wherever C is positive-definite, ensuring gradients propagate K_{VRH} or G_{VRH} back to update the model parameters.

- **Thermal expansion coefficient:** The thermal expansion coefficient relates to thermodynamic quantities through the Grüneisen parameter γ and temperature-dependent bulk modulus [25]:

$$\alpha(T) = \frac{\gamma C_v(T)}{K_T(T) V_{\text{molar}}} \quad (12)$$

where $C_v(T)$ is the heat capacity from the Debye model and the bulk modulus varies with temperature according to:

$$K_T(T) = K_0(1 - \alpha_K T) \quad (13)$$

The linear thermal expansion follows:

$$\frac{\Delta L}{L_0} = \int_{T_0}^T \alpha(T') dT' \quad (14)$$

The model predicts the zero-temperature bulk modulus K_0 , its temperature coefficient α_K , and the Grüneisen parameter γ . Since all operations are smooth and the thermal expansion coefficient depends differentiably on these predicted quantities, gradients flow seamlessly from experimental thermal expansion measurements back to optimize the model parameters for representation learning.

Together, these examples illustrate the breadth of our framework: CLOUD-pDOS captures full vibrational spectra for C_v and S_{vib} , VRH averaging provides smooth and bounded mappings for elastic modulus, and thermodynamic relations yield temperature-dependent thermal expansion. These cases highlight the broader promise of combining CLOUD with physical laws wherever sufficient training data are available.

Reviewer #2 Comments

Comment: *This work introduces a transformer-based framework for crystal representation learning. The representation is SCOPE, a coordinate-free crystal string that concatenates space-group generator symbols (rather than just space group numbers),*

occupied Wyckoff positions, and composition information. The model is pretrained on OPTIMADE structures and fine-tuned on a wide suite of downstream materials property benchmarks (MatBench, MatBench-Discovery, UnconvBench). The reported results show a broadly competitive performance compared to graph neural networks (GNNs) and better than composition-based models. Attention analysis suggest the model attends strongly to symmetry tokens, supporting interpretability claims. The authors also investigate the scaling-law fits related to the data/model size to performance, offering guidance for future scaling. The authors further present a CLOUD-DEBYE that predicts a Debye temperature and propagates it through a differentiable Debye model to obtain heat capacity and internal energy curves, improving accuracy while enforcing thermodynamic consistency. Overall, the work is comprehensive and timely, covering symmetry-aware representations, foundation models, and physics-informed ML for materials. However, the work can be better organized to show better novelty and clarity. I suggest that authors have a revision before the work can be accepted.

Author Reply: We sincerely thank the reviewer for the comprehensive and constructive evaluation of our work. We are grateful for the recognition of the breadth and timeliness of our study, including the development of the SCOPE representation, the CLOUD and CLOUD-DEBYE models, and the exploration of interpretability and scaling laws. We also appreciate the reviewer’s thoughtful suggestion regarding organization and clarity, and we have carefully revised the manuscript to better highlight the novelty of our work and improve the overall presentation.

Comment 1: *One key innovation in this study is the replacement of space group numbers with string generator with symmetry operations. The concept seems appealing and makes sense intuitively, but an ablation study is missing. It is unclear how much gain comes from (a) generator-level symmetry tokens vs space group numbers, (b) inclusion of Wyckoff positions. Comparisons to MatInFormer are confounded by dataset size, tokenizer scope, and model capacity differences.*

Author Reply: We thank the reviewer for the comment. We agree that the design choice of our SCOPE representation is intuitively plausible while we can provide more empirical evidence to support the claim. We also agree that the direct comparison to MatInFormer, though partly suggesting the benefit of using generator strings compared to space group numbers or Hermann-Mauguin notations, is not comprehensive enough given the different pre-training data and model capacity. Therefore, we have designed different variants of SCOPE:

- Varying space group notations: index (1-230), Hermann-Mauguin notations, or Generator strings.
- Inclusion of Wyckoff positions (WPs): using WPs in SCOPE or not.

In addition, we experimented with another variant which has composition-only for comparison, excluding space group and Wyckoff position information. Then we pre-train the model with all those variants and fine-tune them on two representative datasets from MatBench [19]: `dielectric` and `e form`, which have 4764 and 132752 data points respectively. We use the empirical result to demonstrate the effectiveness of using generator strings as the space group notations and the inclusion of Wyckoff positions for improved model predictive accuracy.

We have pre-trained CLOUD for each of the SCOPE variants and fine-tuned the models on `dielectric` and `e form` datasets. As shown in Table 3, across both datasets, including WPs consistently improves performance, regardless of the space group notation used. Meanwhile, generator strings with WPs achieve the best performance, yielding the lowest MAE on both `dielectric` (0.3038) and `e form` (0.0542). Using generator strings without WPs ranks second on `dielectric`, suggesting that generator-level symmetry tokens alone provide a tangible gain over index or HM notations. The composition-only variant performs worst, confirming the value of structural symmetry information. These results quantitatively validate the benefit of both generator-level symmetry tokens and the inclusion of WPs, with their combination offering the largest improvement.

All the above results have been merged into the section S3.7 Ablation Studies in the revised Supplementary Information, which is attached after the response to Comment 2.

Comment 2: *“Long-range order/interactions” claim is underspecified and under-validated. The manuscript attributes CLOUD’s advantage on UnconvBench and phonon thermodynamics to capturing long-range effects but provides no operational definition or ablation test. Prior work shows that mainstream GNNs often fail to capture periodicity and long-range information. The statement will be much stronger if the impact of the string generator of space group numbers on disordered materials. It will be nice if the authors can conduct a controlled ablation study on (a) SG-number only, (b) SG Hermann–Mauguin tokens, (c) Full SCOPE w/o Wyckoff, (d) Symmetry-only (mask composition), (e) Composition-only for perfect and disordered crystals datasets.*

Author Reply: We thank the reviewer for the comment. To address the concern, we have conducted a controlled ablation study on different space group notations (index,

Hermann–Mauguin, generator string) and on the inclusion or exclusion of Wyckoff positions, similar to the ablation studies described in our response to Comment 1. We did not include the symmetry-only variant in the ablation study, though suggested by the reviewer, as symmetry without composition does not define a physically meaningful material and thus would not provide an interpretable baseline. We benchmarked these variants of SCOPE on two datasets: the **defected** dataset from UnconvBench [26], which evaluates the representation’s ability to model defected materials, and the C_v dataset from Gong et al. [27], which probes the capacity to capture long-range interactions. The experimental results are summarized in Table 3 and demonstrate the relative contributions of different symmetry encodings and the inclusion of Wyckoff positions.

For the **defected** dataset, we observe that the SCOPE variants based on the generator string outperform those based on space group index or Hermann–Mauguin notation, indicating that the generator string indeed provides a richer encoding of symmetry. Interestingly, the inclusion of Wyckoff positions slightly degrades performance on defected structures. This trend is likely due to how the dataset was constructed: vacancies were introduced by removing atoms with unique Wyckoff positions [28], but the specific defect site is not annotated. As a result, including Wyckoff positions may introduce noise rather than additional information in this task. For the C_v dataset, the composition-only baseline performs poorly, confirming that composition alone is insufficient to model the structure-property relationship. However, we see a substantial improvement once symmetry information is incorporated, regardless of the specific notation, and the generator string yields the best overall results. This demonstrates that symmetry encodings play a central role in enabling the model to capture thermodynamic properties linked to long-range interactions. Overall, these results support our design choice of using generator strings while highlighting potential limitations when Wyckoff information is incomplete or ambiguous.

Combining our response to Comment 1 & 2 from the reviewer, we have added the following results and discussion regarding the ablations studies in the section S3.7 Ablation Studies in the Supplementary Information:

To systematically evaluate the design choices underlying our SCOPE representation, we conduct a series of ablation studies across multiple datasets. Specifically, we vary (i) the notation used to encode space group symmetry—numerical index (1–230), Hermann–Mauguin (HM) symbols, or generator strings (GS) of symmetry operations—and (ii) whether Wyckoff positions (WPs) are included. We also examined a composition-only

baseline, where symmetry information is ignored entirely, to assess the added value of structural encodings. For each variant, we pre-train CLOUD and fine-tune the resulting models on four representative benchmark datasets: **dielectric** and **e form** from MatBench [19], the **defected** dataset from UnconvBench [26], and the C_v dataset from Gong et al. [27]. These tasks span both conventional property prediction benchmarks and settings where long-range interactions or structural disorder are especially relevant.

The results, summarized in Table 3, yield several consistent trends. First, across the MatBench tasks, including WPs consistently improves accuracy regardless of the space group notation, with generator strings plus WPs achieving the best performance overall. This indicates that both generator-level tokens and fine-grained Wyckoff position information are crucial in modeling the material structures. Meanwhile, for the **defected** dataset, generator strings again provide the strongest symmetry encoding, but the inclusion of WPs slightly reduces performance. This can be attributed to the dataset construction: vacancies are introduced at unique Wyckoff sites without explicit annotation of defect location [28], making WP tokens a source of noise rather than signal. Finally, in the C_v task, the composition-only baseline performs poorly, underscoring the insufficiency of stoichiometry alone for material representation learning. Incorporating symmetry information markedly improves accuracy, with generator strings delivering the lowest errors. Together, these findings quantitatively validate the benefit of generator-level symmetry encoding while also clarifying the contexts in which Wyckoff positions provide complementary information.

Table 3: Ablation study on the SCOPE representation across four datasets with different variants of SCOPE design. Results are reported as the mean test MAE for **dielectric**, **e form**, and **defected** and MAE/MAD for C_v (and standard deviation) from five-fold cross-validation. Bold indicates the best performance within each dataset, and underline denotes the second-best.

SG notation	WP	dielectric (\downarrow)	e form (\downarrow)	defected (\downarrow)	C_v (\downarrow)
N/A	X	0.3438 (0.0916)	0.0845 (0.0007)	0.8544 (0.1894)	0.48 (0.01)
Index	X	0.3551 (0.0974)	0.0599 (0.0006)	0.7769 (0.1946)	0.17 (0.00)
	\checkmark	0.3345 (0.0806)	0.0568 (0.0004)	0.9360 (0.1897)	0.17 (0.01)
HM	X	0.3333 (0.0751)	0.0604 (0.0003)	0.8188 (0.1890)	0.18 (0.00)
	\checkmark	0.3290 (0.0894)	0.0569 (0.0008)	0.8257 (0.1539)	0.16 (0.01)
GS	X	0.3289 (0.0771)	0.0601 (0.0006)	0.7482 (0.1648)	0.17 (0.01)
	\checkmark	0.3038 (0.0782)	0.0542 (0.0007)	0.8197 (0.1093)	0.16 (0.00)

Comment 3: Distinguish “long-range order” (periodicity) from “long-range interactions” (physical couplings such as elastic or electrostatic); the manuscript alternates between them.

Author Reply: We thank the reviewer for this constructive suggestion. We are sorry for the confusion due to the inappropriate use of “long-range order” and “long-range interactions”. We have corrected our terminology throughout the manuscript to clearly distinguish between long-range order (periodicity) — the translational symmetry and repeating arrangement of atoms in the crystal lattice — and long-range interactions — the physical couplings, such as elastic or electrostatic effects, whose influence extends over many interatomic distances. Specifically, we now use long-range order when referring to the periodic arrangement encoded by the crystal’s symmetry, and long-range interactions when referring to the property-relevant couplings that our model captures.

We have corrected the usage of terminologies in the revised manuscript accordingly:

- In Introduction, the original text was ‘Figure 1d) and assess its ability to predict phonon internal energy and heat capacity, two properties that rely on long-range dependencies of crystal structures.’ We have changed ‘dependencies’ to ‘interactions’.
- In the Transferable and scalable crystal representation learning with CLOUD section in Results, the original text was ‘... enabling it to effectively models long-range dependencies and contextual relationships within sequences.’ We have changed ‘dependencies’ to ‘interactions’.
- In the CLOUD-DEBYE: Integration of CLOUD in a differentiable and physics-consistent materials thermodynamics framework section in Results, the original text was ‘... , which restrict their receptive fields and impair their ability to learn long-range dependencies.’ We have changed ‘dependencies’ to ‘orders’.
- In the CLOUD-DEBYE: Integration of CLOUD in a differentiable and physics-consistent materials thermodynamics framework section in Results, the original text was ‘Unlike GNNs, they capture interactions between all tokens without relying on sequential message passing, thereby enabling the modeling of long-range dependencies.’ We have changed ‘dependencies’ to ‘orders’.
- In the CLOUD-DEBYE: Integration of CLOUD in a differentiable and physics-consistent materials thermodynamics framework section in Results, the original text was ‘The results manifest CLOUD’s capability of learning long-range dependencies in crystal structures as well as the significance of introducing thermodynamic consistency into the material machine learning framework.’ We have changed ‘dependencies’ to ‘interactions’.

- In Discussion, the original text was ‘Moreover, CLOUD is capable of learning long-range structural dependencies.’ We have changed ‘dependencies’ to ‘information’.

Comment 4: *Clarify de-duplication rule (“smallest volume per formula + SG”) and report how many items dropped.*

Author Reply: We thank the reviewer for pointing out the need for clarification. Our deduplication method, adapted from Antunes et al. [13], groups structures by identical chemical composition and space group number, and retains only the entry with the smallest volume per formula unit in each group. This approach, which preferentially selects the most compact configuration, removes approximately 51.5% of entries from the raw dataset, leaving ~6.3 M structures for pre-training. The revised manuscript now includes this description and the corresponding statistics in the Methods section for clarity.

We have added the following clarification in the first paragraph of the Datasets section in **Methods**:

To remove duplicate entries, we apply the deduplication rule from Antunes et al. [13]: structures are first grouped by identical chemical composition and space group number, and within each group we retain only the structure with the smallest volume per formula unit. This criterion follows the observation that structures with identical composition and space group but larger volumes are often redundant polymorphs or duplicates of less compact configurations. Applying this rule to our dataset removes ~51.5% of the total structures, resulting in ~6.3M structures for subsequent processing.

Comment 5: *Some relevant work are not cited. A few generative symmetry-aware representations e.g. WyCryst, WyFormer, and SymmCD—demonstrate the importance of explicit symmetry and Wyckoff encodings for both generation and property prediction.*

Author Reply: We thank the reviewer for the comment. We acknowledge that the reviewer mentioned very important and inspiring literature which are highly relevant to our work. Therefore, we have added discussion about the importance of explicit symmetry and Wyckoff encodings and cited the relevant work in our revised manuscript.

We have added the following discussion into the revised manuscript, which could be found in the first and last paragraph in the Symmetry-Consistent Ordered Parameter Encoding (SCOPE) crystal representation section in **Results**:

“Meanwhile, recent work has demonstrated the importance of explicitly incorporating crystallographic symmetry and Wyckoff information into generative and predictive models. For example, WyCryst [29] introduces a Wyckoff-based representation that preserves symmetry during generation and relaxation; WyFormer [30] encodes crystals as fused element–Wyckoff tokens to enable symmetry-conditioned autoregressive generation; SymmCD [31] decomposes crystals into asymmetric units and symmetry operations, ensuring symmetry-preserving diffusion even for rare space groups. These works underscore that symmetry-aware, Wyckoff-based encodings serve as effective and expressive representations for crystal structures, substantially improving the physical validity and diversity of generated crystals.”

“This provides a more granular and physically complete encoding than categorical site-symmetry tokens [30] or binary symmetry matrices [31]. Unlike WyCryst’s one-hot representation with degrees of freedom [29], SCOPE captures the complete set of symmetry generators in a compact symbolic form, enabling fine-grained distinction between closely related space groups and naturally aligning with masked language modeling for large-scale transformer pretraining.”

Reviewer #3 Comments

Comment: *The authors present a framework for predicting the properties of crystalline materials and compare it with multiple deep learning models, which is a hot topic. However, I have some concerns about the novelty of the framework and the performance comparison. The lack of application of some models in specific material systems also means that there is a lack of proof of the effectiveness of the models in practical applications.*

Author Reply: We sincerely thank the reviewer for the thoughtful assessment of our work and for recognizing the relevance of crystal property prediction as a timely and important research direction. We appreciate the reviewer’s concerns regarding the novelty of our framework, the performance comparisons, and the demonstration of practical applications. We have carefully addressed these points in our revision by clarifying the unique contributions of our approach, providing additional comparisons to various baselines, and expanding the discussion of applications to better support the effectiveness of the proposed model.

Comment 1: *The authors perform deduplication on the OPTIMADE dataset, but more details are not shown.*

Author Reply: We thank the reviewer for pointing out the need for clarification. The OPTIMADE dataset indeed contains duplicate structures that can arise from multiple database sources reporting the same material. To address this, we adopted the deduplication strategy from Antunes et al. [13], which identifies duplicates by grouping structures with the same chemical composition and space group number, then retaining only the one with the smallest volume per formula unit. This step removed approximately 51.5% of entries from the raw OPTIMADE collection, yielding ~6.3 M unique structures for training.

We have added the following clarification in the first paragraph of the Datasets section in **Methods**:

To remove duplicate entries, we apply the deduplication rule from Antunes et al. [13]: structures are first grouped by identical chemical composition and space group number, and within each group we retain only the structure with the smallest volume per formula unit. This criterion follows the observation that structures with identical composition and space group but larger volumes are often redundant polymorphs or duplicates of less compact configurations. Applying this rule to our dataset removes ~51.5% of the total structures, resulting in ~6.3M structures for subsequent processing.

Comment 2: *The authors pre-trained on the OPTIMADE dataset, evaluated the performance on the MatBench dataset, and compared it with other deep learning models (CGCNN, ALIGNN). Have CGCNN and ALIGNN also been pre-trained? If not, the performance comparison is unfair.*

Author Reply: We thank the reviewer for the comment. To clarify, the results by CGCNN and ALIGNN reported in our manuscript are not from pre-trained models but trained from scratch in the supervised learning approach instead. While pre-training has become a well-established paradigm for transformer-based language models due to their natural compatibility with self-supervised tasks (e.g., masked language modeling which is used for pre-training CLOUD) and scalability across millions of unlabeled structures, large-scale self-supervised pre-training for crystal GNNs remains non-trivial.

We emphasize two reasons why this comparison remains fair. Most available pre-trained GNNs, such as ALIGNN, rely on supervised pre-training with labeled properties (e.g., energy or force labels, or transferring a model trained for one property to a similar property), which is fundamentally different from our self-supervised approach

and requires costly labeled data. Second, recent attempts to bring self-supervised pre-training to GNNs, such as Crystal Twins [32], have shown only modest improvements over supervised baselines. Crystal Twins adopts a twin-network framework with CGCNN encoders, using data augmentations (random perturbations, atom masking, and edge masking) and objectives such as Barlow Twins or SimSiam losses. While the method shows some performance gains over a purely supervised CGCNN, these improvements are not substantial, nor consistent across all benchmark datasets, and remain far from surpassing state-of-the-art models such as ALIGNN. This highlights both the promise and the current difficulty of making GNN pre-training competitive at scale.

To further probe this issue, we also conducted self-supervised pre-training of CGCNN using the same ~ 6.3 M crystal structures employed for CLOUD, in contrast to the 428K one used in the Crystal Twins paper [32]. In practice, the training proved prohibitively slow with the large dataset: after eight hours of computation on NVIDIA H100 GPU, only 12% of the pre-training steps of one epoch were completed, whereas CLOUD requires on the order of ten minutes per epoch under comparable conditions. This dramatic efficiency gap underscores the scalability advantage of transformer-based models for large-scale pre-training. We acknowledge that our implementation may not be fully optimized—whether due to our own unfamiliarity with the Crystal Twins framework or limitations in the current codebase. Nevertheless, the evidence indicates that self-supervised pre-training for GNNs remains far from mature or practically effective at scale. Indeed, even in the Crystal Twins public repository (<https://github.com/RishikeshMagar/Crystal-Twins/issues/2>), the authors note that attempting ALIGNN pre-training leads to prohibitive compute and memory requirements due to the construction of additional line graphs, and that their preliminary experiments on smaller datasets yielded negligible benefits.

We further support this conclusion with quantitative results on the MatBench GVRH dataset, summarized in Table 4. Crystal Twins yields only minor improvements for CGCNN (from 0.089 to ~ 0.086 – 0.087), and our own large-scale pre-training results in even negative gains, probably because of the under-optimization. In contrast, CLOUD shows a clear and consistent benefit from pre-training ($0.117 \rightarrow 0.087$), with improvements that are both statistically significant and comparable in magnitude to or exceeding those of Crystal Twins. Even though the accuracy by pre-trained CGCNN and CLOUD is comparable, CGCNN requires atomic coordinates as input while our representation is coordinate-free, highlighting the advantage of CLOUD in effective representation learning. Note that `gvrh` is one of the few tasks from MatBench that

CLOUD is not significantly outperforming the non-pretrained or pre-trained CGCNN. As a matter of fact, based on the results from Crystal Twins paper [32], CLOUD outperforms the pre-trained CGCNN on four out of the eight regression tasks while even the non-pretrained CLOUD outperforms the pre-trained CGCNN on two tasks.

Taken together, these findings suggest that while GNNs are strong supervised learners, their current pre-training strategies are less scalable and less effective compared to transformer-based language models like CLOUD. We therefore believe that the comparison is fair and meaningful, as it reflects the state of the art in each modeling paradigm.

We have provided the following results and discussions in the section S3.8 Training Efficiency and Scalability in the revised Supplementary Information:

For baseline comparisons, we follow the MatBench protocol where CGCNN and ALIGNN are trained from scratch in a supervised manner, without pre-training. While ALIGNN provides pre-trained checkpoints, these are obtained via supervised pre-training on labeled data such as energies and forces, in contrast to the self-supervised strategy adopted for CLOUD. Attempts to develop self-supervised pre-training for GNNs, such as the Crystal Twins framework [32], have demonstrated only modest and inconsistent improvements over supervised CGCNN and remain less effective than ALIGNN. We further apply the Crystal Twins framework to pre-train CGCNN on the same $\sim 6.3\text{M}$ crystal structures as CLOUD, but find that the pre-training process is prohibitively slow: after eight hours, only $\sim 12\%$ of the training steps in one epoch are completed, compared to ~ 10 minutes per epoch for CLOUD. We acknowledge that our implementation may not be fully optimized, but the evidence nevertheless indicates that self-supervised pre-training for GNNs is still far from mature. The comparative results on the MatBench GVRH dataset (Table 4) further support this point: while CGCNN with Crystal Twins pre-training achieves only marginal gains over its supervised counterpart, CLOUD demonstrates a substantial and consistent improvement when pre-trained. These observations highlight both the limited effectiveness and poor scalability of GNN-based pre-training, underscoring why transformer-based models like CLOUD are particularly well-suited for large-scale self-supervised learning with crystals.

Comment 3: *Large-scale pre-training is very expensive. Have the authors considered the issue of computational time? training time, testing time, model parameters.*

Author Reply: We thank the reviewer for the comment. For the results reported in the manuscript, we performed pre-training on the deduplicated OPTIMADE dataset ($\sim 6.3\text{M}$ sequences). On two NVIDIA H100 GPUs, the base CLOUD model ($\sim 110\text{M}$ parameters) required only ~ 10 minutes per epoch, which is a reasonable cost compared

Table 4: Comparison of model performance on the MatBench gyvrh dataset. Results are reported as the mean of five-fold MAE with standard deviations. Results for CGCNN (supervised) and Crystal Twins variants are taken from [32]; CLOUD and replicated CGCNN-CT results are from this work.

Model	Pre-training	MAE \pm std
CGCNN [32]	None (supervised)	0.089 \pm 0.001
CGCNN (Crystal Twins) [32]	SSL (CTBarlow)	0.086 \pm 0.004
CGCNN (this work)	SSL (6.3M, CT code)	0.092 \pm 0.004
CLOUD (this work)	None (supervised)	0.117 \pm 0.003
CLOUD (this work)	SSL (6.3M, MLM)	0.087 \pm 0.003

to other large-scale machine learning models for materials science. More importantly, large-scale pre-training with CLOUD is computationally efficient due to the excellent scalability of transformer-based language models. With access to larger computing resources, the wall-clock training time can be further reduced with minimal loss of training efficiency. Indeed, benchmarking on Aurora, the exascale supercomputer at Argonne National Laboratory, confirmed that CLOUD maintains high parallel efficiency up to 64 nodes, even without specialized adaptations for large-scale compute. The results will be added to our revised manuscript. Once pretrained, the model can be reused across all fine-tuning tasks, thereby amortizing the one-time pre-training cost. As the number of downstream applications increases, the relative expense of pre-training becomes almost negligible.

To further assess computational efficiency in practical settings, we benchmark fine-tuning and inference costs relative to common GNN baselines on two representative benchmarks (Jarvis and MatBench FE). Table 5 shows that CLOUD achieves shorter training times per epoch and significantly faster inference, despite its larger model parameter count. These results demonstrate that CLOUD is not only scalable in pre-training but also computationally efficient and lightweight in downstream deployment, indicating its practicality for large-scale discovery pipelines.

Furthermore, compared to graph-based models that require explicit 3D atomic coordinates or costly DFT relaxations, CLOUD substantially lowers the end-to-end computational burden for large-scale discovery workflows. Most state-of-the-art models for material structure modeling are based on graph neural networks (GNNs), which face intrinsic scaling challenges such as oversmoothing and oversquashing [33]. In addition, distributed training of GNNs requires graph partitioning, where boundary nodes

and edges induce significant cross-device communication overhead. In contrast, transformers benefit from well-established parallelization strategies that scale efficiently with modern hardware.

Overall, while pre-training CLOUD requires a non-trivial but one-time investment, the efficiency of training at scale, and reusability across tasks, justifies the cost. We therefore view pre-training as a worthwhile tradeoff for enabling efficient and scalable downstream applications in materials science tasks.

We have added the following results and discussions in the section S3.8 Training Efficiency and Scalability in the revised Supplementary Information:

The computational cost of large-scale pre-training is an important consideration for practical application of foundation models. For the results reported in this work, we pre-train CLOUD on the deduplicated OPTIMADE dataset (~ 6.3 M sequences). Using two NVIDIA H100 GPUs, the base model (~ 110 M parameters) requires ~ 10 minutes per epoch. Such a cost is modest compared to other large-scale machine learning models in materials science and is further mitigated by the fact that pre-training is a one-time investment. Once obtained, a pretrained checkpoint can be reused across diverse fine-tuning tasks, consequently amortizing the initial cost over a broad range of downstream applications.

To evaluate the scaling efficiency of CLOUD on the Aurora HPC, the exascale supercomputer at Argonne National Laboratory, we measure the model throughput while scaling the number of nodes for two versions of the CLOUD model: a 110M parameter baseline and a 1B parameter larger model. The weak scaling setup fixes the local batch size per GPU, allowing us to isolate the effect of communication overhead as the cluster size grows. As shown in Figure 6a, both models sustain high efficiency up to 128 nodes, maintaining efficiency higher than 60%. Notably, the 1B model consistently outperforms the 110M model in scalability, particularly at higher node counts, benefiting from a larger compute footprint that better amortizes communication overhead. We evaluate the strong scaling behavior of the CLOUD model on Aurora for the two versions of model. In this setup, the total training workload remains constant as the number of nodes increases, and we record the reduction in runtime to compute speedup. As shown in Figure 6b, the 110M model shows near-linear speedup up to 64 nodes and reasonable scaling beyond. The 1B model, evaluated from 32 to 128 nodes, demonstrates consistent gains, though with diminishing returns at higher scales due to communication overhead. The log-log plots highlight the efficiency of distributed training, validating our stack’s readiness for exascale foundation model training.

Furthermore, compared to graph-based models that require explicit 3D atomic coordinates or costly DFT relaxations, CLOUD substantially lowers the end-to-end computational burden for large-scale discovery workflows. Most state-of-the-art models for material

(a) Scaling Efficiency vs. Number of Nodes for training CLOUD with 110M and 1B parameter models on Aurora using our training stack. Efficiency is computed under weak scaling by fixing the local batch size per GPU. Both models maintain high parallel efficiency up to 128 nodes (higher than 60%), with the 1B model exhibiting better scalability at large node counts due to improved compute-to-communication ratio.

(b) Scaling Speedup vs. Number of Nodes for training CLOUD with 110M and 1B parameter models on Aurora using our training stack. Speedup is computed relative to the smallest measured node count (8 for 110M, 32 for 1B), with ideal scaling shown as dashed lines. Both the two models manifest near-perfect strong scaling behavior up to 64 nodes for 110M model or 128 nodes for 1B model.

Fig. 6: Scalability analysis for CLOUD on Aurora [34].

structure modeling are based on graph neural networks (GNNs), which face intrinsic scaling challenges such as oversmoothing and oversquashing [33]. In addition, distributed training of GNNs requires graph partitioning, where boundary nodes and edges induce significant cross-device communication overhead. In contrast, transformers benefit from well-established parallelization strategies that scale efficiently with modern hardware.

In addition to pre-training efficiency, the practicality of CLOUD also depends on the cost of fine-tuning and inference. To further assess computational efficiency in practical settings, we compared training and inference times of CLOUD against widely used GNN baselines (ALIGNN [1], Matformer [35], MEGNet [6], M3GNet [36], TensorNet [37], and SO3Net [38]) on representative property prediction tasks (Jarvis [39] and MatBench [19] formation energy datasets). The results for ALIGNN and Matformer on Jarvis formation energy task are obtained from Yan et al. [35] while the numbers for the MatBench formation energy task are collected from implementations with MatGL [40]. As summarized in Table 5, CLOUD achieves faster training time per epoch and significantly shorter inference time despite being larger in model parameter count. For example, on the Jarvis FE task, CLOUD requires only 34 seconds per epoch and 19 seconds for full inference, outperforming ALIGNN and Matformer while offering an order-of-magnitude faster inference throughput. Similarly, on the MatBench FE benchmark, CLOUD completes inference in only 7 seconds on the test set, the fastest among all tested models. These results highlight

that the transformer-based architecture not only scales efficiently during pre-training but also provides lightweight fine-tuning and inference in downstream applications.

Table 5: Training and inference time comparison on material property prediction tasks. Reported numbers include training time per epoch, inference time on the full dataset (for Jarvis formation energy task) or the test set (for MatBench formation energy task), and model size.

Model	Time/epoch	Inference	Parameters
ALIGNN (Jarvis FE)	327 s	156 s	15.4 M
Matformer (Jarvis FE)	64 s	59 s	11.0 M
CLOUD (Jarvis FE)	34 s	19 s	110 M
MEGNet (MatBench FE)	N/A	11.137 s	168 K
M3GNet (MatBench FE)	N/A	20.089 s	228 K
TensorNet (MatBench FE)	N/A	13.694 s	2.8 M
SO3Net (MatBench FE)	N/A	32.601 s	341 K
CLOUD (MatBench FE)	59.892 s	7.003 s	110 M

Comment 4: *PMCGNN’s performance on the JARVIS-DFT dataset is already higher than ALIGNN, Matformer, etc. The authors should also compare.* <https://pubs.acs.org/doi/10.1021/acs.jcim.4c01200>

Author Reply: We thank the reviewer for the comment. We acknowledge that PMCGNN is indeed an important model with superior performance on many benchmark tasks. Therefore, we have fine-tuned CLOUD on JARVIS-DFT dataset, following the same data splitting strategies as described in Feng et al. [41].

We have fine-tuned CLOUD on the five datasets from JARVIS-DFT: formation energy, bandgap (OPT), total energy, bandgap (MBJ), and E_{hull} . We follow the same dataset version and data-splitting strategy as described in Feng et al. [41] and report the results in Table 6. Our model CLOUD achieves competitive predictive performance, consistently outperforming CGCNN across the five tasks and exhibits the state-of-the-art (SOTA) performance on the E_{hull} dataset. For 4 out of the 5 tasks, PMCGNN stands out with the lowest MAE, reflecting its tailored graph-based inductive bias. Nevertheless, CLOUD reaches comparable accuracy while being coordinate-free and symmetry-consistent by design, highlighting its complementary strengths. In particular, achieving the best performance on E_{hull} demonstrates CLOUD’s capability for stability predictions, which are of central importance for materials discovery.

Table 6: MAE Comparison between CLOUD and PMCGNN as well as other baselines on JARVIS Dataset. The results for CGCNN, ALIGNN, Matformer, and PMCGNN are directly obtained from Feng et al. [41].

Model	formation energy ev/atom	bandgap (OPT) ev	total energy ev/atom	bandgap (MBJ) ev	E_{hull} eV
CGCNN	0.063	0.20	0.078	0.41	0.17
ALIGNN	0.0331	0.142	0.037	0.31	0.076
Matformer	0.0325	0.137	0.035	0.30	0.064
PMCGNN	0.0278	0.122	0.029	0.25	0.040
CLOUD	0.0606	0.159	0.069	0.31	0.030

We have also added the results and related discussion in S3.4 JARVIS-DFT section in the revised Supplementary Information:

Additionally, we have also fine-tuned CLOUD on JARVIS-DFT-2021 3D dataset [39]. We chose five crystal property prediction tasks: formation energy, bandgap (OPT), total energy, bandgap(MBJ), and E_{hull} for benchmarking. We follow the same data-splitting strategy as described in Feng et al. [41], and compare our model performance with the following baselines: CGCNN [5], ALIGNN [1], Matformer [35], and PMCGNN [41]. As shown in Table 6, our model CLOUD achieves competitive predictive performance, consistently outperforming CGCNN across the five tasks and exhibits the state-of-the-art (SOTA) performance on the E_{hull} dataset. For 4 out of the 5 tasks, PMCGNN stands out with the lowest MAE, reflecting its tailored graph-based inductive bias. Nevertheless, CLOUD reaches comparable accuracy while being coordinate-free and symmetry-consistent by design, highlighting its complementary strengths. In particular, achieving the best performance on E_{hull} demonstrates CLOUD’s capability for stability predictions, which are of central importance for materials discovery.

Plus, we have also cited this excellent paper as one of the pioneering work in this field in the second paragraph of the **Introduction** in the revised manuscript:

Pioneering work has significantly improved the prediction accuracy for crystals [1, 5, 41–44].

Comment 5: *The author also mentioned that the performance of the current model depends on a large amount of labeled data, but the author’s pre-training-fine-tuning model also depends on downstream labeled data. The self-supervised model may better handle this problem: CDSSL, <https://doi.org/10.48550/arXiv.2408.17255>.*

Author Reply: We thank the reviewer for the comment. CDSSL [45] is an effective pre-training approach for predictive models, in particular, graph neural networks

(GNNs) with a pretext task based on recovering valid material structures when given perturbed versions of these structures.

Meanwhile, we would like to clarify that CDSSL is, like CLOUD, a pre-train-finetune framework in which labeled data is still required for downstream property prediction. Both methods aim to improve data efficiency when fine-tuning on limited labeled data, rather than eliminating the need for labeled data entirely. In comparison, CDSSL pre-trains structure-based graph neural networks via a geometric denoising objective, while CLOUD pre-trains a transformer-based model on our SCOPE representation using masked language modeling. CDSSL has shown improvements for GNNs by learning local geometric information, whereas CLOUD learns global symmetry and compositional patterns.

In addition, These approaches are complementary, and future work could explore combining SCOPE-based global symmetry encoding with CDSSL-style geometric denoising to enhance data efficiency further.

We have added the citation for the CDSSL paper and included the related discussion in the third paragraph in the **Introduction** section:

Aside from language-model-style pre-training, alternative self-supervised strategies have been explored in the crystal domain, such as Crystal Denoising Self-Supervised Learning (CDSSL) [45], which pre-trains graph neural networks by perturbing atomic positions and learning to recover the original interatomic distances.

References

- [1] Choudhary, K., DeCost, B.: Atomistic line graph neural network for improved materials property predictions. *npj Computational Materials* **7**(1), 185 (2021)
- [2] Riebesell, J., Goodall, R.E., Jain, A., Benner, P., Persson, K.A., Lee, A.A.: Mat-bench discovery—an evaluation framework for machine learning crystal stability prediction. *arXiv preprint arXiv:2308.14920* (2023)
- [3] Jain, A., Ong, S.P., Hautier, G., Chen, W., Richards, W.D., Dacek, S., Cholia, S., Gunter, D., Skinner, D., Ceder, G., et al.: Commentary: The materials project: A materials genome approach to accelerating materials innovation. *APL materials* **1**(1) (2013)
- [4] Wang, H.-C., Botti, S., Marques, M.A.: Predicting stable crystalline compounds

- using chemical similarity. *npj Computational Materials* **7**(1), 12 (2021)
- [5] Xie, T., Grossman, J.C.: Crystal graph convolutional neural networks for an accurate and interpretable prediction of material properties. *Physical review letters* **120**(14), 145301 (2018)
- [6] Chen, C., Ye, W., Zuo, Y., Zheng, C., Ong, S.P.: Graph networks as a universal machine learning framework for molecules and crystals. *Chemistry of Materials* **31**(9), 3564–3572 (2019)
- [7] Grygorenko, O.O., Radchenko, D.S., Dziuba, I., Chuprina, A., Gubina, K.E., Moroz, Y.S.: Generating multibillion chemical space of readily accessible screening compounds. *iScience* **23**(11), 101681 (2020) <https://doi.org/10.1016/j.isci.2020.101681>
- [8] Bellmann, L., Penner, P., Gastreich, M., Rarey, M.: Comparison of combinatorial fragment spaces and its application to ultralarge make-on-demand compound catalogs. *Journal of Chemical Information and Modeling* **62**(3), 553–566 (2022) <https://doi.org/10.1021/acs.jcim.1c01378> <https://doi.org/10.1021/acs.jcim.1c01378>. PMID: 35050621
- [9] Patel, H., Ihlenfeldt, W.-D., Judson, P.N., Moroz, Y.S., Pevzner, Y., Peach, M.L., Delannée, V., Tarasova, N.I., Nicklaus, M.C.: Savi, in silico generation of billions of easily synthesizable compounds through expert-system type rules. *Scientific Data* (2020)
- [10] Gavezzotti, A.: Are crystal structures predictable? *Accounts of chemical research* **27**(10), 309–314 (1994)
- [11] Merchant, A., Batzner, S., Schoenholz, S.S., Aykol, M., Cheon, G., Cubuk, E.D.: Scaling deep learning for materials discovery. *Nature*, 1–6 (2023)
- [12] Zeni, C., Pinsler, R., Zügner, D., Fowler, A., Horton, M., Fu, X., Shysheya, S., Crabbé, J., Sun, L., Smith, J., et al.: Mattergen: a generative model for inorganic materials design. *arXiv preprint arXiv:2312.03687* (2023)
- [13] Antunes, L.M., Butler, K.T., Grau-Crespo, R.: Crystal structure generation with autoregressive large language modeling. *arXiv preprint arXiv:2307.04340* (2023)

- [14] Gruver, N., Sriram, A., Madotto, A., Wilson, A.G., Zitnick, C.L., Ulissi, Z.: Fine-tuned language models generate stable inorganic materials as text. arXiv preprint arXiv:2402.04379 (2024)
- [15] Barroso-Luque, L., Shuaibi, M., Fu, X., Wood, B.M., Dzamba, M., Gao, M., Rizvi, A., Zitnick, C.L., Ulissi, Z.W.: Open materials 2024 (omat24) inorganic materials dataset and models. arXiv preprint arXiv:2410.12771 (2024)
- [16] Deng, B., Zhong, P., Jun, K., Riebesell, J., Han, K., Bartel, C.J., Ceder, G.: Chgnet as a pretrained universal neural network potential for charge-informed atomistic modelling. *Nature Machine Intelligence* **5**(9), 1031–1041 (2023)
- [17] Siron, M., Djafar, I., Fayet, E., Rossello, A., Ramlaoui, A., Duval, A.: Lemat-bulk: aggregating, and de-duplicating quantum chemistry materials databases. In: *AI for Accelerated Materials Design-ICLR 2025* (2025)
- [18] Xu, C., Zhu, S., Viswanathan, V.: Cloud: A scalable scientific foundation model for crystal representation learning. In: *Neurips 2024 Workshop Foundation Models for Science: Progress, Opportunities, and Challenges* (2024)
- [19] Dunn, A., Wang, Q., Ganose, A., Dopp, D., Jain, A.: Benchmarking materials property prediction methods: the matbench test set and automatminer reference algorithm. *npj Computational Materials* **6**(1), 138 (2020)
- [20] Furukawa, G.T., Douglas, T.B., McCoskey, R.E., Ginnings, D.C.: Thermal properties of aluminum oxide from 0 to 1200 k. *Journal of research of the National Bureau of Standards* **57**(2), 67–82 (1956)
- [21] Stull, D.R.: *JANAF Thermochemical Tables... vol. 1*. Clearinghouse, Washington, D.C. (1965)
- [22] Woodfield, B.F., Shapiro, J.L., Stevens, R., Boerio-Goates, J., Putnam, R.L., Helean, K.B., Navrotsky, A.: Molar heat capacity and thermodynamic functions for CaO . *The Journal of Chemical Thermodynamics* **31**(12), 1573–1583 (1999)
- [23] Leitner, J., Strejc, A., Sedmidubsky, D., Ruzicka, K.: High temperature enthalpy and heat capacity of CaO . *Thermochimica acta* **401**(2), 169–173 (2003)
- [24] Loew, A., Sun, D., Wang, H.-C., Botti, S., Marques, M.A.: Universal machine learning interatomic potentials are ready for phonons. arXiv preprint

arXiv:2412.16551 (2024)

- [25] Grüneisen, E.: Theorie des festen zustandes einatomiger elemente. *Annalen der Physik* **344**(12), 257–306 (1912)
- [26] Wang, H., Sun, J., Liang, J., Zhai, L., Tang, Z., Li, Z., Zhai, W., Wang, X., Gao, W., Gong, S.: Crystograph: A comprehensive predictive model for crystal material properties and the benchmark. *Battery Energy*, 70004 (2024)
- [27] Gong, S., Yan, K., Xie, T., Shao-Horn, Y., Gomez-Bombarelli, R., Ji, S., Grossman, J.C.: Examining graph neural networks for crystal structures: limitations and opportunities for capturing periodicity. *Science Advances* **9**(45), 3245 (2023)
- [28] Choudhary, K., Sumpter, B.G.: Can a deep-learning model make fast predictions of vacancy formation in diverse materials? *AIP Advances* **13**(9) (2023)
- [29] Zhu, R., Nong, W., Yamazaki, S., Hippalgaonkar, K.: Wycryst: Wyckoff inorganic crystal generator framework. *Matter* **7**(10), 3469–3488 (2024)
- [30] Kazeev, N., Nong, W., Romanov, I., Zhu, R., Ustyuzhanin, A., Yamazaki, S., Hippalgaonkar, K.: Wyckoff transformer: Generation of symmetric crystals. arXiv preprint arXiv:2503.02407 (2025)
- [31] Levy, D., Panigrahi, S.S., Kaba, S.-O., Zhu, Q., Lee, K.L.K., Galkin, M., Miret, S., Ravanbakhsh, S.: Symmcd: Symmetry-preserving crystal generation with diffusion models. arXiv preprint arXiv:2502.03638 (2025)
- [32] Magar, R., Wang, Y., Barati Farimani, A.: Crystal twins: self-supervised learning for crystalline material property prediction. *npj Computational Materials* **8**(1), 231 (2022)
- [33] Wang, Y., Cho, K.: Non-convolutional graph neural networks. arXiv preprint arXiv:2408.00165 (2024)
- [34] Argonne Leadership Computing Facility: Aurora. <https://www.alcf.anl.gov/aurora>
- [35] Yan, K., Liu, Y., Lin, Y., Ji, S.: Periodic graph transformers for crystal material property prediction. *Advances in Neural Information Processing Systems* **35**, 15066–15080 (2022)

- [36] Chen, C., Ong, S.P.: A universal graph deep learning interatomic potential for the periodic table. *Nature Computational Science* **2**(11), 718–728 (2022)
- [37] Simeon, G., De Fabritiis, G.: TensorNet: Cartesian tensor representations for efficient learning of molecular potentials. *Advances in Neural Information Processing Systems* **36**, 37334–37353 (2023)
- [38] Schütt, K.T., Hessmann, S.S., Gebauer, N.W., Lederer, J., Gastegger, M.: Schnetpack 2.0: A neural network toolbox for atomistic machine learning. *The Journal of Chemical Physics* **158**(14) (2023)
- [39] Choudhary, K., Garrity, K.F., Reid, A.C., DeCost, B., Biacchi, A.J., Hight Walker, A.R., Trautt, Z., Hattrick-Simpers, J., Kusne, A.G., Centrone, A., *et al.*: The joint automated repository for various integrated simulations (jarvis) for data-driven materials design. *npj computational materials* **6**(1), 173 (2020)
- [40] Ko, T.W., Deng, B., Nassar, M., Barroso-Luque, L., Liu, R., Qi, J., Thakur, A.C., Mishra, A.R., Liu, E., Ceder, G., *et al.*: Materials graph library (matgl), an open-source graph deep learning library for materials science and chemistry. *npj Computational Materials* **11**(1), 253 (2025)
- [41] Feng, H., Tian, H.: Improving crystal property prediction from a multiplex graph perspective. *Journal of Chemical Information and Modeling* **64**(19), 7376–7385 (2024)
- [42] Gilmer, J., Schoenholz, S.S., Riley, P.F., Vinyals, O., Dahl, G.E.: Neural message passing for quantum chemistry. In: *International Conference on Machine Learning*, pp. 1263–1272 (2017). PMLR
- [43] Goodall, R.E., Lee, A.A.: Predicting materials properties without crystal structure: Deep representation learning from stoichiometry. *Nature communications* **11**(1), 6280 (2020)
- [44] Ruff, R., Reiser, P., Stühmer, J., Friederich, P.: Connectivity optimized nested graph networks for crystal structures. *arXiv preprint arXiv:2302.14102* (2023)
- [45] New, A., Le, N.Q., Pekala, M.J., Stiles, C.D.: Self-supervised learning for crystal property prediction via denoising. *arXiv preprint arXiv:2408.17255* (2024)

Response to Reviews on Article “CLOUD: A
Scalable and Physics-Informed Foundation Model
for Crystal Representation Learning”

Changwen Xu¹, Shang Zhu¹, Venkatasubramanian Viswanathan^{1,2*}

¹Department of Mechanical Engineering, University of Michigan.

²Department of Aerospace Engineering, University of Michigan.

*Corresponding author(s). E-mail(s): venkvis@umich.edu;
Contributing authors: changwex@umich.edu; shangzhu@umich.edu;

Manuscript ID: NCOMMS-25-45609

We are very grateful for the reviewers’ comments, suggestions, and valuable input on our manuscript. We are pleased that the previous round of revisions has addressed most of the concerns. In what follows, we would like to answer the remaining questions. We strongly believe that the revised manuscript should meet the standards for publication in Nature Communications.

Reviewer #1 Comments

Comment: *The authors have addressed my comments properly, and I suggest acceptance of this paper to be published in Nature Communications.*

Author Reply: We appreciate that our previous response has addressed the reviewer's comments. We also sincerely thank the reviewer for the constructive feedback, which has helped improve the quality and clarity of our manuscript. We further thank the reviewer for recommending acceptance of our paper for publication in Nature Communications.

Reviewer #2 Comments

Comment: *The authors did a good job in addressing my comments. I don't have further questions.*

Author Reply: We are glad that our previous response has satisfactorily addressed the reviewer's concerns. We are grateful to the reviewer for the constructive suggestions, which have strengthened the clarity and overall quality of our manuscript.

Reviewer #3 Comments

Comment: *I thank the authors for including additional model comparisons and validation analyses, which help support the model's performance and are central to this study's motivation. That said, my earlier concern remains: the proposed CLOUD method does not consistently outperform PMCGNN across key metrics. According to Table 6, only Ehull (reported in eV) shows a strong result, with other measures lacking clear improvement. To strengthen the case for CLOUD, I encourage the authors to*

highlight its distinct advantages, whether in terms of scalability, robustness, or applicability in specific scenarios, to better demonstrate its unique value beyond the current benchmarks.

Author Reply: We sincerely thank the reviewer for the thoughtful comment and for recognizing that the additional analyses we provided strengthen the paper’s motivation. We agree that CLOUD does not consistently surpass PMCGNN across all five JARVIS tasks. PMCGNN is a strong state-of-the-art GNN baseline, and our results show that while CLOUD does not dominate it uniformly, CLOUD consistently outperforms CGCNN and achieves performance comparable to other widely used baselines such as ALIGNN across these benchmarks.

At the same time, we appreciate the reviewer’s suggestion to highlight CLOUD’s distinct advantages beyond benchmark accuracy. We would like to clarify that the core goal of this work is not merely to introduce another benchmark competitor, but rather to demonstrate how a scalable scientific foundation model can bridge machine learning and domain physics for materials discovery. Therefore, the revised manuscript already emphasizes several unique strengths of CLOUD:

- **Scalability.** In the section “Transferable and scalable crystal representation learning with CLOUD” in Results, we report scaling analyses showing that CLOUD maintains strong performance with respect to dataset size and model size. In our prior response letter for the last round of revision, we also provided scaling plots from HPC cluster runs illustrating CLOUD’s favorable scaling behavior.
- **Inference efficiency.** As shown in Table 5 of our previous response letter, CLOUD achieves faster inference than graph neural networks such as ALIGNN, Matformer, MEGNet, M3GNet, TensorNet, and SO3Net—even though CLOUD contains more parameters. This advantage is central for large-scale screening workflows.
- **Integration with differentiable physics.** A major contribution of this work is CLOUD-DEBYE, which integrates CLOUD with the Debye model in a differentiable framework. With training data of heat capacity only under 300K, CLOUD-DEBYE predicts temperature-dependent $C_v(T)$ accurately for different temperatures and in a physics-consistent manner. These results are detailed in the section “CLOUD-DEBYE: Integration of CLOUD in a differentiable and physics-consistent materials thermodynamics framework” in Results.
- **Practical materials discovery potential.** In the last revision, we expanded the evaluation to WBM dataset from MatBench Discovery, adding convex-hull-based screening metrics and a case study of Al_2Fe . These results highlight CLOUD’s applicability to real discovery workflows beyond standard supervised benchmarks.

Overall, these evidence show that CLOUD provides distinct advantages in scalability, inference speed, physics integration, and real-world applicability, even if it does not uniformly outperform PMCGNN on all supervised metrics. We appreciate the reviewer's guidance in clarifying CLOUD's unique values.